# Global credit assignment via dynamical criticality

**Wentao Wang** [1 2 3]  **Keren Gao** [1]  **Guozhang Chen** [1]

## Abstract

Efficiently training recurrent neural networks on long sequences remains an open challenge. The standard global paradigm, backpropagation through time (BPTT), suffers from vanishing and exploding gradients and memory costs that scale linearly with sequence length. Conversely, biologically inspired local learning rules are memory-efficient but typically introduce severe bias. To bridge this gap, we introduce Criticality-driven Online Local Alignment (COLA). By leveraging the long-range spatiotemporal correlations inherent to the critical regime, COLA enables a strictly local learning rule to approximate global error propagation, thereby combining online efficiency with gradient descent precision. Theoretically, for a recurrent neural network with $H$ hidden units, COLA requires only an $\mathcal{O}(H)$ auxiliary state and constant activation memory, completely independent of sequence length. Empirically, COLA is competitive with BPTT on standard benchmarks and demonstrates superior robustness on stability-sensitive tasks. Finally, we conduct a rigorous analysis of the approximation error to provide a theoretical foundation for reliable online learning. Code is available at github.com/Criticality-Cognitive-Computation-Lab/COLA.

## 1. Introduction

Many intelligent systems must extract causal structure from temporal experience to predict future outcomes and guide decision-making (Sutton et al., 1998; Friston, 2010). In recurrent models, this requires solving temporal credit assignment: determining how parameters acting at time $t$

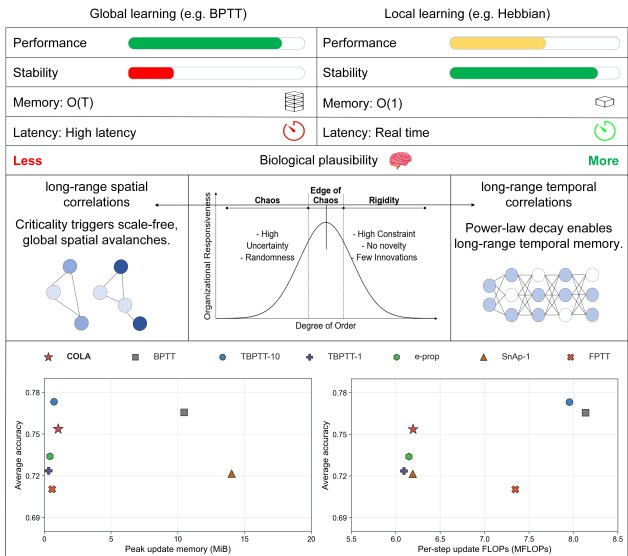

*Figure 1.* Comparison of credit assignment paradigms. Global BPTT is accurate but memory-intensive ($\mathcal{O}(T)$), while local rules are efficient ($\mathcal{O}(1)$) but less precise. COLA bridges this gap by leveraging long-range spatiotemporal correlations in a stable near-critical regime to approximate global gradients with constant memory, achieving a favorable trade-off between accuracy, memory, and cost.

should be updated using errors that may only emerge many steps later (Rumelhart et al., 1986; Richards et al., 2019). Backpropagation through time (BPTT) (Werbos, 1990) remains the standard solution because it differentiates the exact supervised objective. However, exactness comes with two well-known costs: activation memory grows linearly with sequence length $T$, and the backward signal depends on long Jacobian products that are prone to vanishing or exploding (Bengio et al., 1994; Pascanu et al., 2013). As a result, long-horizon learning remains expensive, latency-heavy, and often fragile. The resulting gap between high-fidelity gradients and biologically plausible online computation remains a central challenge (Whittington & Bogacz, 2019; Lillicrap et al., 2020).

Figure 1 illustrates the resulting trade-off between gradient fidelity and online efficiency. In the bottom panels, the cost coordinates are computed from per-step profiler outputs on the common row-sequential classification regime (Row-MNIST, Row-CIFAR10, UCI HAR). Prior work has

[1]State Key Laboratory of Multimedia Information Processing and the National Engineering Research Center of Visual Technology, School of Computer Science, Peking University, Beijing, China [2]Dalian University of Technology, Dalian, China [3]Supervised by Guozhang Chen. Correspondence to: Guozhang Chen <guozhang.chen@pku.edu.cn>.

*Proceedings of the 43rd International Conference on Machine Learning*, Seoul, South Korea. PMLR 306, 2026. Copyright 2026 by the author(s).

attacked this bottleneck along two largely separate directions. Architectural mechanisms such as LSTM (Hochreiter & Schmidhuber, 1997) and GRU (Cho et al., 2014) improve gradient transport through gated or additive state dynamics, but they are still typically optimized with BPTT and therefore do not remove the need for full-trajectory storage. Exact online differentiation via real-time recurrent learning (RTRL) (Williams & Zipser, 1989) eliminates temporal unrolling at the price of prohibitive high-order sensitivity tensors. Approximate online alternatives, including compressed RTRL variants, eligibility-trace rules, and forward-mode surrogates (Tallec & Ollivier, 2017; Mujika et al., 2018; Menick et al., 2021; Murray, 2019; Bellec et al., 2020; Kag & Saligrama, 2021), improve tractability but typically pay with synapse-scale memory, higher-variance or biased gradients, or surrogate objectives. The central challenge is therefore not merely to make learning online, but to do so without retaining global, parameter-scale sensitivity state.

Increasing evidence from neuroscience suggests that cortical dynamics typically operate near criticality and that the brain exhibits long-range spatiotemporal correlations in this regime. In particular, neuronal avalanches and scale-invariant coordinated activity in cortex reveal critical features across spatial scales (Beggs & Plenz, 2003; Petermann et al., 2009; Shew et al., 2009; 2015). Meanwhile, long-range temporal correlations in EEG/MEG amplitude fluctuations indicate that brain activity likewise exhibits nontrivial persistence across time scales (Linkenkaer-Hansen et al., 2001). Inspired by this picture, we place the network in a stable near-critical state and model the long-range spatial and temporal correlations that emerge in this regime to directly approximate the BPTT teaching signal.

We make the following contributions in this work. **An online learning rule:** Criticality-driven Online Local Alignment (COLA) approximates BPTT gradients in closed form using only within-step quantities, reducing the computational cost without sacrificing much performance. **Dynamical criticality as an enabler:** COLA leverages near-critical network dynamics (edge of chaos) to justify two key assumptions—temporal persistence and long-range spatial correlation of gradient signals—enabling accurate scalarization. **Constant memory and linear time complexity:** COLA achieves $\mathcal{O}(1)$ activation memory independent of sequence length $T$, requiring only an $\mathcal{O}(H)$ extra state in a Recurrent Neural Network (RNN) with $H$ hidden states.

## 2. Related Work

**BPTT, truncation, and architectural mechanisms.** Backpropagation through time (BPTT) (Werbos, 1990) remains the standard solution for training recurrent models, yet it suffers from two major drawbacks. First, its memory cost scales linearly with sequence length $T$. Second, the

backward signal depends on long Jacobian products that are notoriously prone to vanishing or exploding gradients (Bengio et al., 1994; Pascanu et al., 2013), which hinders the learning of long-horizon dependencies. To mitigate these costs, Truncated BPTT (TBPTT) (Williams & Peng, 1990) limits the backward pass to a fixed window; however, this introduces gradient bias and fails to capture dependencies beyond the truncation horizon. While architectural mechanisms like LSTM (Hochreiter & Schmidhuber, 1997) and GRU (Cho et al., 2014) improve gradient transport through gated state dynamics, they are still typically optimized via temporal unrolling. Consequently, they do not resolve the fundamental offline credit-assignment bottleneck or the associated memory requirements. COLA targets the same supervised objective but circumvents these issues by replacing the backward recursion with a forward-only local approximation.

**Online learning alternatives.** Online alternatives differ primarily in what they approximate and the volume of auxiliary state they maintain. RTRL (Williams & Zipser, 1989) computes exact gradients forward in time, but its influence tensor incurs prohibitive $\mathcal{O}(H^4)$ computational complexity and $\mathcal{O}(H^3)$ state storage. Approximate-RTRL methods, such as UORO (Tallec & Ollivier, 2017), Kronecker-factored RTRL (Mujika et al., 2018), and SnAp (Menick et al., 2021), compress this tensor through stochastic low-rank factors or sparsity; although they reduce the cost, they still retain parameter-scale auxiliary variables, which amount to $\mathcal{O}(H^2)$ for a dense RNN with $H$ hidden units. A parallel line of biologically inspired work, including RFLO (Murray, 2019) and e-prop (Bellec et al., 2020), combines local plasticity with global error modulation via random feedback or eligibility traces. Similarly, forward-mode approaches such as FPTT (Kag & Saligrama, 2021) avoid reverse-time passes by solving local subproblems instead of the exact gradient of the original loss. While these rules operate online, they often exhibit performance gaps and still necessitate the storage of $\mathcal{O}(H^2)$ auxiliary states. In contrast, COLA directly approximates the BPTT teaching signal and retains the original supervised objective, reducing the auxiliary state to merely $\mathcal{O}(H)$ by summarizing recurrent feedback with per-unit scalar statistics.

**Dynamics, criticality, and stability.** From a dynamical-systems perspective, learning quality is closely tied to whether recurrent dynamics are contractive, near-critical, or chaotic. Random recurrent networks undergo a transition to chaos as gain increases (Sompolinsky et al., 1988), and computational performance has often been argued to peak near the edge of chaos (Bertschinger et al., 2004). Empirical neuroscience offers a related but more nuanced picture: long-range temporal correlations in EEG/MEG (Linkenkaer-Hansen et al., 2001), neuronal avalanches in cortex (Beggs

& Plenz, 2003; Petermann et al., 2009), maximal dynamic range near criticality (Shew et al., 2009), and adaptation toward criticality in visual cortex (Shew et al., 2015) collectively motivate cortical operating regimes close to criticality, while reverberating-network analyses suggest that in vivo activity typically remains on the stable, slightly subcritical side (Wilting et al., 2018). In this work, criticality serves only to specify the operating regime of interest, rather than to assert that cortical dynamics are tuned to an exact critical point. On the machine-learning side, FORCE (Sussillo & Abbott, 2009), predictive alignment (Asabuki & Clopath, 2025), feedback-control views of recurrent credit assignment (Kaleb et al., 2024), and Lyapunov-guided analyses (Vogt et al., 2024) likewise emphasize the role of dynamical regime in trainability. Our use of Lyapunov diagnostics is closest in spirit to this line, and is more direct in purpose: to initialize a stable near-critical working band in which COLA's scalar closure is empirically reliable.

## 3. Methods

This section derives the COLA learning rule starting from dynamical preconditions. We first define the RNN dynamics and local derivatives in Sec. 3.1. We then place the model in a stable near-critical initialization band using an operator-normalized Lyapunov diagnostic and a direct gain estimator (Sec. 3.2). In this regime, recurrent signals exhibit enough long-range temporal correlations and long-range spatial correlations to support the scalar closure used by COLA. These properties allow us to approximate the BPTT recursion by a same-step system, yielding the closed-form teaching signal in Secs. 3.3.2–3.3.3. After presenting the online estimators for the required statistics in Sec. 3.3.4, we analyze the resulting computational and memory complexity in Sec. 3.4. Finally, we defer architectural extensions to Appendices J, K, and L.

**Notation.** We index time steps by $t \in \{1, \dots, T\}$. Let $H, D$, and $K$ denote the hidden state dimension, input dimension, and output dimension, respectively. Throughout this paper, vectors are denoted by bold lowercase letters ($\mathbf{x}$) and matrices by bold uppercase letters ($\mathbf{W}$); $\mathbf{I}_H$ represents the $H \times H$ identity matrix. We use $\odot$ and $\oslash$ to denote element-wise (Hadamard) multiplication and division, respectively. For any element-wise nonlinearity $\phi(\cdot)$, $\frac{\partial \mathbf{h}}{\partial \mathbf{x}}$ denotes the vector of its first derivatives. Asymptotic complexities are expressed using standard notation $\mathcal{O}(\cdot)$.

### 3.1. Model and local derivatives

We consider a standard RNN architecture. Let $\mathbf{u}_t \in \mathbb{R}^D$ denote the input, $\mathbf{x}_t \in \mathbb{R}^H$ the pre-activation, and $\mathbf{h}_t \in \mathbb{R}^H$ the hidden state at time $t$. Assuming an elementwise nonlinearity $\phi(\cdot)$ (typically $\tanh$), the system evolves according

to the following recurrence and readout equations:

$$\mathbf{x}_t = \mathbf{W}_{hh}\mathbf{h}_{t-1} + \mathbf{W}_{xh}\mathbf{u}_t + \mathbf{b}_h, \qquad (1)$$
$$\mathbf{h}_t = \phi(\mathbf{x}_t),$$
$$\mathbf{o}_t = \mathbf{W}_{hy}\mathbf{h}_t + \mathbf{b}_y, \qquad (2)$$
$$\mathbf{p}_t = \mathrm{softmax}(\mathbf{o}_t),$$

where $\mathbf{W}_{hh} \in \mathbb{R}^{H \times H}$, $\mathbf{W}_{xh} \in \mathbb{R}^{H \times D}$, and $\mathbf{W}_{hy} \in \mathbb{R}^{K \times H}$ denote the recurrent, input, and readout weight matrices, respectively.

The training objective is to minimize a cumulative loss function, defined as the weighted sum of per-step losses:

$$\mathcal{L}(\theta) = \sum_{t=1}^{T} \mathcal{L}_t = \sum_{t=1}^{T} \bar{w}_t \, \ell(\mathbf{o}_t, \mathbf{y}_t^\star), \qquad (3)$$

where $\theta$ denotes all trainable parameters, $\mathbf{y}_t^\star \in \mathbb{R}^K$ is the target at time $t$ (e.g., one-hot for classification), $\bar{w}_t \geq 0$ is an optional per-step weight (default $\bar{w}_t{=}1$), and $\ell(\cdot)$ represents the specific loss function, such as cross-entropy for classification or squared error for regression.

For the standard softmax cross-entropy case, the gradient of the instantaneous loss with respect to the logits $\mathbf{o}_t$ has a concise closed-form solution. Considering that $\mathbf{p}_t = \mathrm{softmax}(\mathbf{o}_t)$ represents the probability distribution predicted by the model, this gradient is expressed as:

$$\frac{\partial \mathcal{L}_t}{\partial \mathbf{o}_t} = \bar{w}_t \, (\mathbf{p}_t - \mathbf{y}_t^\star). \qquad (4)$$

By applying the chain rule through the readout layer and the elementwise nonlinearity, we obtain the local pre-activation gradient:

$$\frac{\partial \mathcal{L}_t}{\partial \mathbf{x}_t} = \frac{\partial \mathbf{h}_t}{\partial \mathbf{x}_t} \odot \mathbf{W}_{hy}^\top \frac{\partial \mathcal{L}_t}{\partial \mathbf{o}_t}. \qquad (5)$$

This term represents the immediate error signal available at step $t$ before accounting for temporal dependencies.

### 3.2. Task-aware near-critical initialization

COLA does not require the network to remain exactly critical throughout training. What it needs is a stable starting regime with sufficiently long recurrent echoes for the local closure to be informative. We therefore use Lyapunov diagnostics only as an initialization scaffold that places the recurrent dynamics near the stable boundary.

The local stability of the hidden state trajectories is governed by the instantaneous Jacobian:

$$\mathbf{J}_t \triangleq \frac{\partial \mathbf{h}_t}{\partial \mathbf{h}_{t-1}} = \mathrm{diag}(\frac{\partial \mathbf{h}_t}{\partial \mathbf{x}_t})\mathbf{W}_{hh}. \qquad (6)$$

To quantify the global dynamical regime, we compute the maximum Lyapunov exponent $\lambda_{\max}$, which represents the

average exponential growth rate of infinitesimal perturbations. Following the standard QR algorithm applied to products of Jacobians along data-driven trajectories (Skokos, 2009) $\lambda_{\max}$ is approximated as:

$$\lambda_{\max} \approx \frac{1}{T} \log \sigma_{\max}\left(\prod_{t=1}^{T} \mathbf{J}_t\right). \qquad (7)$$

A value of $\lambda_{\max} > 0$ indicates chaotic sensitivity, while $\lambda_{\max} < 0$ implies contraction; our target is the critical transition point where $\lambda_{\max} \approx 0$.

To make gains comparable across tasks and random initializations, we factor the recurrent operator into a normalized shape and a scalar gain:

$$\mathbf{W}_{hh}(g) = g\,\overline{\mathbf{W}}_{hh}. \qquad (8)$$

$$\overline{\mathbf{W}}_{hh} = \frac{\mathbf{W}_{hh}}{\|\mathbf{W}_{hh}\|_{\mathrm{op}}}, \qquad \text{with} \qquad \|\overline{\mathbf{W}}_{hh}\|_{\mathrm{op}} = 1. \quad (9)$$

Empirically, near the stable boundary the maximum Lyapunov exponent is close to affine in $\log g$:

$$\lambda_{\max}(g) \approx a + b\log g. \qquad (10)$$

The key point is that after operator normalization the explicit gain factor contributes one unit of slope in $\log g$, while the remaining deviation of $b$ away from 1 comes only from how the driven contraction term changes with gain. This yields two direct root estimates with different cost–accuracy trade-offs. The cheaper unit estimate uses only the single probe at $g=1$ and treats the residual slope correction as locally negligible, i.e. $b \approx 1$:

$$g_{\mathrm{unit}} = \exp\bigl(-\lambda_{\max}(1)\bigr). \qquad (11)$$

This is a one-point tangent approximation to the root of the pre-training Lyapunov curve. It is cheapest, but it becomes biased when the task-dependent contraction changes appreciably with gain. The more accurate two-point estimate adds one nearby stable-side probe $g_2 > 1$ and replaces the fixed unit slope by an empirical secant slope:

$$\widehat{b} = \frac{\lambda_{\max}(g_2) - \lambda_{\max}(1)}{\log g_2},$$
$$g_{\mathrm{two}} = \exp\Bigl(-\frac{\lambda_{\max}(1)}{\widehat{b}}\Bigr). \qquad (12)$$

Equivalently, the two-point estimator corrects the local slope mismatch induced by nonlinear activation slopes, leak/adaptation terms, and the state distribution seen by the task. It requires only one extra Lyapunov evaluation and is exact whenever $\lambda_{\max}$ is affine in $\log g$ on the probed interval. Appendix H reports task-level performance for the unit estimator, the two-point estimator, and multi-gain sweep initialization across the evaluated tasks.

## 3.3. Proposed method: Criticality-driven Online Local Alignment (COLA)

For the COLA learning rule, we begin by formulating the exact gradient recursion used in BPTT, and then introduce two key hypotheses regarding signal propagation in the near-critical regime: long-range temporal correlations and spatial mode alignment. By applying these dynamical assumptions to the exact BPTT recursion, we derive the strictly online, closed-form COLA update rule. Finally, we analyze the time, activation memory, and extra state of all methods evaluated in the subsequent section.

### 3.3.1. BPTT BASELINE

BPTT computes the exact total gradient with respect to pre-activations

$$\boldsymbol{\delta}_t \triangleq \frac{\partial \mathcal{L}}{\partial \mathbf{x}_t}, \qquad (13)$$

which satisfies the backward recursion

$$\boldsymbol{\delta}_t = \frac{\partial \mathcal{L}_t}{\partial \mathbf{x}_t} + \frac{\partial \mathbf{h}_t}{\partial \mathbf{x}_t} \odot \bigl(\mathbf{W}_{hh}^{\top}\boldsymbol{\delta}_{t+1}\bigr). \qquad (14)$$

The full teaching-signal recursion and notation are derived in Appendix A. Parameter gradients take an outer-product form, e.g.,

$$\frac{\partial \mathcal{L}}{\partial \mathbf{W}_{hh}} = \sum_{t=1}^{T} \boldsymbol{\delta}_t\,\mathbf{h}_{t-1}^{\top}, \qquad (15)$$

with analogous expressions for $\mathbf{W}_{xh}$, $\mathbf{b}_h$, and the readout parameters.

### 3.3.2. TWO NEAR-CRITICAL ASSUMPTIONS

Below we introduce two assumptions, which are verified in Section 4.6.

**Assumption 1 (long-range temporal correlations).** On short time scales in a stable near-critical regime, the BPTT pre-activation gradient $\delta_t^{(i)}$ (Eq. (13)) continues approximately by a per-unit scalar,

$$\delta_{t+1}^{(i)} \approx \alpha_i\,\delta_t^{(i)}, \qquad (16)$$

with slowly varying coefficients $\alpha_i$. Intuitively, near-critical dynamics yield long correlation times, so the backward signal evolves smoothly over short windows.

**Assumption 2 (long-range spatial correlations).** In a stable near-critical regime, the within-step teaching signal is assumed to exhibit long-range spatial correlation. Specifically, for a fixed unit $i$, the error signal of any unit $j$ is approximately proportional to that of unit $i$:

$$\delta_t^{(j)} \approx \beta_{ji}\,\delta_t^{(i)}, \qquad \beta_{ji} \text{ slowly varying in } t.$$

Appendix B.3 shows step-by-step how this proportionality collapses the recurrent summation in Eq. (19) into a per-unit loop gain $\mu_i$ used below.

### 3.3.3. FROM BPTT TO A CLOSED-FORM ONLINE TEACHING SIGNAL

Starting from the BPTT recursion (Eq. (14)) and applying Assumption 1 ($\boldsymbol{\delta}_{t+1} \approx \operatorname{diag}(\boldsymbol{\alpha})\boldsymbol{\delta}_t$), we arrive at a linearized implicit system. Crucially, this formulation eliminates the explicit dependency on future states, yielding a relationship involving only same-step quantities:

$$\boldsymbol{\delta}_t \approx \frac{\partial \mathcal{L}_t}{\partial \mathbf{x}_t} + \mathbf{A}_t \boldsymbol{\delta}_t, \qquad (17)$$

where the effective instantaneous Jacobian $\mathbf{A}_t$ is defined as:

$$\mathbf{A}_t \triangleq \operatorname{diag}\left(\frac{\partial \mathbf{h}_t}{\partial \mathbf{x}_t}\right) \mathbf{W}_{hh}^\top \operatorname{diag}(\boldsymbol{\alpha}). \qquad (18)$$

Appendix B and Appendix B.1 give the step-by-step derivation from the BPTT recursion to this same-step implicit system, with the implicit-system formulation summarized in Appendix B.2.

While this system formally implies a matrix inverse, computing it is prohibitive (A Neumann/path interpretation of the associated resolvent is given in Appendix B.4). To derive a computationally efficient local update, we explicitly expand the matrix-vector product $\mathbf{A}_t \boldsymbol{\delta}_t$ for a single unit $i$. By decomposing $\mathbf{A}_t$ into its constituent scaling and mixing operations, the $i$-th component becomes:

$$\delta_t^{(i)} \approx \frac{\partial \mathcal{L}_t}{\partial x_t^{(i)}} + \frac{\partial h_t^{(i)}}{\partial x_t^{(i)}} \sum_{j=1}^{H} W_{hh}^{ji} \, \alpha_j \, \delta_t^{(j)}. \qquad (19)$$

Applying Assumption 2 directly to the recurrent summation term gives (see Appendix B.3 for the step-by-step derivation)

$$\sum_{j=1}^{H} W_{hh}^{ji} \, \alpha_j \, \delta_t^{(j)} \approx \sum_{j=1}^{H} W_{hh}^{ji} \, \alpha_j \left( \beta_{ji}(t) \, \delta_t^{(i)} \right)$$
$$= \left( \sum_{j=1}^{H} W_{hh}^{ji} \, \alpha_j \, \beta_{ji}(t) \right) \delta_t^{(i)} \equiv \mu_i \, \delta_t^{(i)}, \qquad (20)$$

where $\mu_i$ is the corresponding slowly varying per-unit loop gain. Substituting this closure into Eq. (19) reduces the recurrence to a linear algebraic equation:

$$\delta_t^{(i)} \approx \frac{\partial \mathcal{L}_t}{\partial x_t^{(i)}} + \frac{\partial h_t^{(i)}}{\partial x_t^{(i)}} \mu_i \delta_t^{(i)}. \qquad (21)$$

Solving for $\delta_t^{(i)}$ yields the final closed-form approximation:

$$\widetilde{\delta}_t^{(i)} = \frac{\partial \mathcal{L}_t / \partial x_t^{(i)}}{1 - \mu_i \frac{\partial h_t^{(i)}}{\partial x_t^{(i)}}}. \qquad (22)$$

---

**Algorithm 1** Criticality-aware Online Local Learning (COLA; closed-form, no BPTT)

1: **Input:** Stream data $\{(\mathbf{u}_t, \mathbf{y}_t^\star)\}_{t=1}^T$, global weights $\{\bar{w}_t\}_{t=1}^T$, learning rate $\eta$, EMA rates $\rho_\alpha, \rho_\mu$
2: **Output:** Trained RNN parameters
3: **Initialize** hidden state $\mathbf{h}_0 \leftarrow \mathbf{0}$ and network parameters $\mathbf{W}_{xh}, \mathbf{W}_{hh}, \mathbf{W}_{hy}, \mathbf{b}_h, \mathbf{b}_y$
4: **Initialize** unit statistics: $S_i, Q_i, S_{A^2}^{(i)}, S_{AB}^{(i)} \leftarrow 0$, $\mu_i \leftarrow 0$, and $\widetilde{\delta}_0^{(i)} \leftarrow 0$ for all units $i$
5: **for** $t = 1, 2, \ldots, T$ **do**
6:    **Forward pass:** Compute $(\mathbf{x}_t, \mathbf{h}_t, \mathbf{o}_t, \mathbf{p}_t)$ via Eqs. (1)–(2)
7:    **Local gradients:** Compute $\partial \mathcal{L}_t / \partial \mathbf{o}_t$ (Eq. (4)), $\frac{\partial \mathbf{h}_t}{\partial \mathbf{x}_t}$, and $\partial \mathcal{L}_t / \partial \mathbf{x}_t$ (Eq. (5))
8:    **Teaching signal:** Compute $\widetilde{\boldsymbol{\delta}}_t$ via Eq. (23) with safeguard Eq. (31)
9:    **if** $t \geq 2$ **then**
10:      Update $\alpha_i$ via EMA using $\widetilde{\delta}_t^{(i)}$ and $\widetilde{\delta}_{t-1}^{(i)}$ (Eq. (25))
11:    **end if**
12:    **Parameter update:** Update weights via Eqs. (32)–(33)
13:    **if** $t \geq 2$ **then**
14:      Form $A_t^{(i)}$ and $B_t^{(i)}$ from same-step quantities via Eq. (28)
15:      Update loop gain $\mu_i$ for the next step via Eq. (30)
16:    **end if**
17: **end for**

---

Stacking across all units, the update rule is expressed in vector form as:

$$\widetilde{\boldsymbol{\delta}}_t = \left( \frac{\partial \mathcal{L}_t}{\partial \mathbf{x}_t} \right) \oslash \left( \mathbf{1} - \boldsymbol{\mu} \odot \frac{\partial \mathbf{h}_t}{\partial \mathbf{x}_t} \right), \qquad (23)$$

where $\oslash$ denotes elementwise division. This formulation dramatically reduces the complexity of temporal credit assignment, requiring the maintenance of only two scalars $(\alpha_i, \mu_i)$ per unit.

### 3.3.4. CLOSED-FORM ONLINE ESTIMATION OF $\alpha_i$ AND $\mu_i$

COLA maintains two per-unit scalars: a temporal linearization coefficient $\alpha_i$ (Assumption 1) and the loop gain $\mu_i$ in Eq. (22). Both are estimated online with exponential moving averages.

To estimate $\alpha_i$, we treat the teaching signal as a local autoregressive (AR(1)) model, $\widetilde{\delta}_t^{(i)} \approx \alpha_i \widetilde{\delta}_{t-1}^{(i)}$, and use the least-squares solution

$$\alpha_i^\star = \frac{\mathbb{E}[\widetilde{\delta}_t^{(i)} \widetilde{\delta}_{t-1}^{(i)}]}{\mathbb{E}[(\widetilde{\delta}_{t-1}^{(i)})^2]}, \qquad (24)$$

The least-squares AR(1) estimator is derived in Appendix D,

and we implement it with exponential moving-average (EMA) statistics:

$$S_i(t) \leftarrow \rho_\alpha S_i(t-1) + (1-\rho_\alpha)\, \widetilde{\delta}_t^{(i)} \widetilde{\delta}_{t-1}^{(i)},$$

$$Q_i(t) \leftarrow \rho_\alpha Q_i(t-1) + (1-\rho_\alpha)\, (\widetilde{\delta}_{t-1}^{(i)})^2,$$

$$\alpha_i(t) \leftarrow \mathrm{clip}\left( \frac{S_i(t)}{Q_i(t) + \varepsilon_\alpha}, \alpha_{\min}, \alpha_{\max} \right), \quad (25)$$

with $|\alpha_{\max}| < 1$ to promote stability.

To estimate $\mu_i$, we use Assumption 1 as an identification criterion for the closed-form surrogate: among the candidate loop gains, we prefer the one for which the surrogate teaching signal follows the same short-window continuation law as the target BPTT teaching signal:

$$\widetilde{\delta}_t^{(i)} \approx \alpha_i\, \widetilde{\delta}_{t-1}^{(i)}. \quad (26)$$

Appendix C discusses the role of this temporal self-consistency condition in more detail. Substituting Eq. (22) into Eq. (26) yields a linear relation

$$\mu_i A_t^{(i)} \approx B_t^{(i)}, \quad (27)$$

where

$$A_t^{(i)} \triangleq \alpha_i \frac{\partial h_t^{(i)}}{\partial x_t^{(i)}} \frac{\partial \mathcal{L}_{t-1}}{\partial x_{t-1}^{(i)}} - \phi'(x_{t-1}^{(i)}) \frac{\partial \mathcal{L}_t}{\partial x_t^{(i)}},$$

$$B_t^{(i)} \triangleq \alpha_i \frac{\partial \mathcal{L}_{t-1}}{\partial x_{t-1}^{(i)}} - \frac{\partial \mathcal{L}_t}{\partial x_t^{(i)}}. \quad (28)$$

With exponentially decaying weights $\omega_\tau \propto \rho_\mu^{t-\tau}$, weighted least squares yields

$$\mu_i(t) = \frac{\sum_{\tau \leq t} \omega_\tau A_\tau^{(i)} B_\tau^{(i)}}{\sum_{\tau \leq t} \omega_\tau (A_\tau^{(i)})^2 + \varepsilon_\mu}. \quad (29)$$

Eq. (29) is implemented with two EMA accumulators:

$$S_{A^2}^{(i)}(t) \leftarrow \rho_\mu S_{A^2}^{(i)}(t-1) + (1-\rho_\mu)(A_t^{(i)})^2,$$

$$S_{AB}^{(i)}(t) \leftarrow \rho_\mu S_{AB}^{(i)}(t-1) + (1-\rho_\mu)A_t^{(i)} B_t^{(i)},$$

$$\mu_i(t) \leftarrow \Pi\left[ \frac{S_{AB}^{(i)}(t)}{S_{A^2}^{(i)}(t) + \varepsilon_\mu} \right], \quad (30)$$

Appendix E and Appendix F provide the closed-form one-step and least-squares derivations for $\mu_i$. The projection $\Pi[\cdot]$ enforces a safe denominator, e.g.,

$$\left| \mu_i \frac{\partial h_t^{(i)}}{\partial x_t^{(i)}} \right| \leq 1 - \varepsilon_d. \quad (31)$$

The complete training procedure, integrating the statistic updates and parameter adjustments, is summarized in Algorithm 1.

### 3.3.5. ONLINE PARAMETER UPDATES

Given $\widetilde{\boldsymbol{\delta}}_t$, COLA updates recurrent parameters online with the same outer-product structure as BPTT:

$$\Delta \mathbf{W}_{hh} = -\eta\, \widetilde{\boldsymbol{\delta}}_t \mathbf{h}_{t-1}^\top, \quad (32)$$

$$\Delta \mathbf{W}_{xh} = -\eta\, \widetilde{\boldsymbol{\delta}}_t \mathbf{u}_t^\top,$$

$$\Delta \mathbf{b}_h = -\eta\, \widetilde{\boldsymbol{\delta}}_t.$$

The readout parameters use the exact instantaneous gradient:

$$\Delta \mathbf{W}_{hy} = -\eta \left( \frac{\partial \mathcal{L}_t}{\partial \mathbf{o}_t} \right) \mathbf{h}_t^\top, \quad (33)$$

$$\Delta \mathbf{b}_y = -\eta \frac{\partial \mathcal{L}_t}{\partial \mathbf{o}_t}.$$

Optionally, standard norm clipping is applied to each update tensor.

**Extensions and scope.** Appendix J extends COLA to convolutional RNNs (ConvRNNs) by treating each hidden channel as a unit and maintaining per-channel scalar statistics. Appendix K applies the same principle to spiking recurrent cells whose global recurrence still flows through one coupled state chain, such as RLIF, RadLIF, and EGRU. In contrast, LSTM involves multi-state recurrent dynamics that couple multiple long-lived memory chains, and its results are reported as a pilot study in Appendix L.

### 3.4. Computational and memory complexity

To quantify the efficiency gains of COLA, Table 1 summarizes the asymptotic complexity of various credit assignment methods for a sequence of length $T$. Standard BPTT requires storing the entire history of activations ($\mathcal{O}(TH)$), resulting in a memory footprint that grows linearly with $T$. Although TBPTT mitigates this, it sacrifices exactness across long horizons. Exact online alternatives like RTRL eliminate the $\mathcal{O}(T)$ memory requirement but introduce a prohibitive $\mathcal{O}(H^3)$ state expansion and $\mathcal{O}(TH^4)$ computational cost.

Crucially, while other efficient online approximations (e.g., e-prop, FPTT) maintain recurrent-dominated linear time complexity ($\mathcal{O}(TH^2)$), they typically require storing additional eligibility traces or shadow variables at the parameter scale ($\mathcal{O}(H^2)$). In contrast, because COLA tracks temporal dependencies through compact scalar statistics ($\alpha_i, \mu_i$) as introduced in Sec. 3.3.4, it circumvents both the $\mathcal{O}(T)$ memory bottleneck of BPTT and the $\mathcal{O}(H^2)$ extra state overhead of prior online methods. By requiring only $\mathcal{O}(H)$ extra state and $\mathcal{O}(1)$ activation memory, COLA provides a highly scalable online learning mechanism.

*Table 1.* Asymptotic costs for a vanilla RNN. Here, $T$ denotes the sequence length and $H$ the number of hidden units. We report the recurrent-dominated scaling and suppress input/readout terms shared across methods. "Activation memory" refers to the storage required for hidden states that scales with the sequence length.

| Method | Time (per sequence) | Activation memory | Extra state (excluding parameters) |
|---|---|---|---|
| Full BPTT (Werbos, 1990) | $\mathcal{O}(TH^2)$ | $\mathcal{O}(TH)$ | none |
| TBPTT (truncation $\tau$) | $\mathcal{O}(TH^2)$ | $\mathcal{O}(\tau H)$ | none |
| RTRL (Williams & Zipser, 1989) | $\mathcal{O}(TH^4)$ | $\mathcal{O}(1)$ | $\mathcal{O}(H^3)$ |
| UORO (Tallec & Ollivier, 2017) | $\mathcal{O}(TH^2)$ | $\mathcal{O}(1)$ | rank-one factors $\mathcal{O}(H^2)$ |
| SnAp-1 (Menick et al., 2021) | $\mathcal{O}(TH^2)$ | $\mathcal{O}(1)$ | one-step sparse influences $\mathcal{O}(H^2)$ |
| e-prop (Bellec et al., 2020) | $\mathcal{O}(TH^2)$ | $\mathcal{O}(1)$ | eligibility traces $\mathcal{O}(H^2)$ |
| FPTT (Kag & Saligrama, 2021) | $\mathcal{O}(TH^2)$ | $\mathcal{O}(1)$ | shadow parameter copy $\mathcal{O}(H^2)$ |
| **Ours (COLA)** | $\mathcal{O}(TH^2)$ | $\mathcal{O}(1)$ | $\alpha, \mu$ stats $\mathcal{O}(H)$ |

## 4. Experiments

We evaluate COLA on vanilla RNNs, ConvRNNs, Spiking Neural Networks (SNNs) and LSTMs. To isolate the effect of credit assignment, architectures and task budgets are matched across methods. All main-text tables report mean $\pm$ standard deviation over three seeds (42/43/44). Additional learning curves and supplementary comparisons are collected in Appendix L.

### 4.1. Setup and baselines

**Implementation details.** Unless stated otherwise, we use stochastic gradient descent with a learning rate of $10^{-3}$ and a gradient norm clipping threshold of 5.0. Task-specific batch sizes, training budgets, and data splits are detailed in Appendix Section I.

**COLA configuration.** For COLA, we set the EMA decay rates to $\rho_\alpha{=}0.995$ and $\rho_\mu{=}0.98$, clip $\alpha_i$ to $[-0.99, 0.99]$, and project $\mu_i$ so that $\big|\mu_i \frac{\partial h_t^{(i)}}{\partial x_t^{(i)}}\big| \leq 0.99$ with denominator floor $10^{-3}$. The benchmark tables in the main paper use a shared scan-selected initialization: a short COLA scan chooses the recurrent gain once on the validation split, and the resulting initial parameters are then copied to every baseline. This equalizes the starting point across methods, but it is not a method-specific tuning budget; Appendix L therefore adds tuned-BPTT fairness checks. Appendix H shows that the direct normalized estimators of Sec. 3.2, especially the two-point estimator, recover similar task performance. Appendix L also includes mild sensitivity to $\rho_\alpha$ and the Lyapunov window around these defaults, together with one-time initialization overheads, tuned-BPTT fairness checks, fast-target stress tests, an LSTM pilot, and a long-horizon gesture scope probe.

**Architectures and metrics.** Sequence benchmarks employ single-layer tanh RNNs with a linear readout. The main-text SNN summary table is complemented by the cell equations in Appendix K, while ConvRNN refinement se-

tups are detailed in Appendix J. For regression tasks, we minimize the MSE and report rollout MSE as a measure of long-horizon stability. In classification experiments, we optimize cross-entropy loss and evaluate performance using test accuracy derived from the final-step logits.

**Baselines.** We compare COLA against full BPTT (Werbos, 1990), TBPTT (Williams & Peng, 1990) with window sizes $\tau \in \{1, 10\}$, e-prop (Bellec et al., 2020), FPTT (Kag & Saligrama, 2021), and approximate-RTRL baselines UORO (Tallec & Ollivier, 2017) and SnAp-1 (Menick et al., 2021) whenever the architecture is supported. Exact online gradient methods such as RTRL (Williams & Zipser, 1989) are omitted because their $\mathcal{O}(H^4)$ computation is prohibitive on the sequence lengths considered here. For FPTT, we use a strict chunked implementation with 10 chunks and regularization weight $\mu_{\mathrm{FPTT}}{=}1$, together with oracle mixing and a 20-epoch warmup on classification tasks.

### 4.2. RNN tasks.

Vanilla tanh RNNs are evaluated on three families of sequence benchmarks: long-horizon regression and rollout prediction, represented by the Adding problem (Arjovsky et al., 2016) and Lorenz image prediction (Lorenz, 2017); sequence classification, including Row-sequential MNIST (Lecun et al., 1998), Row-sequential CIFAR-10 (Krizhevsky, 2009), and UCI HAR (Anguita et al., 2013); and character-level language modeling using PTB-Char (Marcus et al., 1993) and WikiText-2 Char (Merity et al., 2017). Quantitative results are summarized in Table 2, and comprehensive task definitions and hyperparameters are provided in Appendix Section I.

**Regression and rollout stability.** COLA is strongest on tasks where long-range recurrent credit assignment and dynamical stability dominate the metric. On Adding, COLA reduces the test MSE to $0.0059 \pm 0.0026$, far below BPTT $(0.1426 \pm 0.0326)$ and the other online baselines. The same pattern is even more pronounced on Lorenz rollout prediction, where COLA reaches $0.0058 \pm 0.0003$; UORO is the

*Table 2.* RNN benchmarks (mean $\pm$ std over seeds 42/43/44). Best among the methods shown is bold.

| Task | BPTT | TBPTT-1 | TBPTT-10 | e-prop | FPTT | UORO | SnAp-1 | **COLA** |
|---|---|---|---|---|---|---|---|---|
| Adding | 0.1426 | 0.0863 | 0.0991 | 0.0813 | 0.1356 | 0.1444 | 0.0860 | **0.0059** |
| MSE↓ | $\pm$ 0.0326 | $\pm$ 0.0031 | $\pm$ 0.0013 | $\pm$ 0.0031 | $\pm$ 0.0336 | $\pm$ 0.0150 | $\pm$ 0.0034 | $\pm$ **0.0026** |
| Lorenz | 0.6841 | 0.2019 | 0.6749 | 0.3602 | 0.8038 | 0.1395 | 0.2025 | **0.0058** |
| MSE↓ | $\pm$ 0.0873 | $\pm$ 0.0845 | $\pm$ 0.0163 | $\pm$ 0.0201 | $\pm$ 0.0250 | $\pm$ 0.1072 | $\pm$ 0.0676 | $\pm$ **0.0003** |
| PTB-Char | 2.8695 | 1.8316 | 2.2272 | 2.0852 | 2.3184 | 2.2279 | **1.8279** | 1.8352 |
| Loss↓ | $\pm$ 0.0203 | $\pm$ 0.0040 | $\pm$ 0.0029 | $\pm$ 0.0013 | $\pm$ 0.0057 | $\pm$ 0.0063 | $\pm$ **0.0041** | $\pm$ 0.0056 |
| WikiText-2 | 3.3620 | 2.2833 | 2.8780 | 2.5851 | 3.0499 | 2.6225 | **2.2793** | 2.4174 |
| Loss↓ | $\pm$ 0.0165 | $\pm$ 0.0058 | $\pm$ 0.0201 | $\pm$ 0.0073 | $\pm$ 0.0307 | $\pm$ 0.0653 | $\pm$ **0.0054** | $\pm$ 0.0035 |
| Row-MNIST | 0.9713 | 0.9343 | **0.9734** | 0.9565 | 0.9481 | 0.3060 | 0.9403 | 0.9544 |
| Acc↑ | $\pm$ 0.0007 | $\pm$ 0.0011 | $\pm$ **0.0020** | $\pm$ 0.0005 | $\pm$ 0.0037 | $\pm$ 0.0193 | $\pm$ 0.0047 | $\pm$ 0.0036 |
| Row-CIFAR10 | 0.4545 | 0.3894 | **0.4759** | 0.3836 | 0.4021 | 0.2185 | 0.3825 | 0.4418 |
| Acc↑ | $\pm$ 0.0108 | $\pm$ 0.0162 | $\pm$ **0.0130** | $\pm$ 0.0026 | $\pm$ 0.0064 | $\pm$ 0.0057 | $\pm$ 0.0070 | $\pm$ 0.0083 |
| UCI HAR | **0.8877** | 0.8416 | 0.8799 | 0.8314 | 0.8367 | 0.5594 | 0.8420 | 0.8690 |
| Acc↑ | $\pm$ **0.0454** | $\pm$ 0.0291 | $\pm$ 0.0390 | $\pm$ 0.0446 | $\pm$ 0.0171 | $\pm$ 0.0359 | $\pm$ 0.0134 | $\pm$ 0.0257 |

*Table 3.* Spiking recurrent benchmarks on directly comparable cell-task pairings. Best within each row is bold.

| Dataset | Cell | BPTT | pp-prop | COLA |
|---|---|---|---|---|
| N-MNIST | RLIF | 98.33% $\pm$ 0.04% | 98.25% $\pm$ 0.03% | **98.54%** $\pm$ **0.03%** |
| N-MNIST | RadLIF | 98.29% $\pm$ 0.02% | **98.40%** $\pm$ **0.03%** | 93.56% $\pm$ 0.12% |
| SHD | RLIF | 94.01% $\pm$ 0.12% | 93.93% $\pm$ 0.28% | **94.07%** $\pm$ **0.45%** |
| SHD | RadLIF | 94.72% $\pm$ 1.06% | 95.33% $\pm$ 0.11% | **95.37%** $\pm$ **0.13%** |
| DVS-Gesture | EGRU | **97.45%** $\pm$ **0.27%** | 97.29% $\pm$ 0.16% | 97.19% $\pm$ 0.32% |

closest non-COLA baseline at $0.1395 \pm 0.1072$, and all other non-COLA baselines remain at or above about $0.20$. These are the settings in which bypassing the long backward chain appears most beneficial.

**Classification.** On row-sequential vision and UCI HAR, COLA is no longer the top method: TBPTT-10 or full BPTT remain strongest, indicating that some tasks still benefit from more exact multi-step gradient transport. Even so, COLA remains competitive with e-prop and FPTT, beats UORO on all three tasks, and stays above SnAp-1 on all three classification benchmarks while trailing BPTT on UCI HAR. This is the correct scope of the claim: COLA is not uniformly better than BPTT, but it remains a viable online alternative when unrolling is undesirable.

**Language modeling.** On PTB-Char and WikiText-2 Char, COLA substantially improves over full BPTT, e-prop, FPTT, and UORO while staying close to the strongest truncated baselines. On PTB, COLA obtains $1.8352 \pm 0.0056$, essentially matching TBPTT-1 ($1.8316 \pm 0.0040$) and trailing SnAp-1 only slightly. On WikiText-2, COLA again im-

proves over BPTT, e-prop, FPTT, and UORO, although SnAp-1 and TBPTT-1 remain stronger.

Furthermore, we provide detailed comparisons of computational costs, including total wall-clock time, MFLOPs, and per-step latency, against approximate RTRL methods in Appendix L. All benchmarks were conducted on a vGPU-32 instance.

### 4.3. SNN tasks

We also evaluate COLA on recurrent SNNs using RLIF, RadLIF, and EGRU cells (Wang et al., 2026). Table 3 summarizes the directly comparable pairings, while Appendix K gives the full aligned benchmark table and the cell equations. Across these controlled comparisons, COLA is competitive in four of the five pairings. On N-MNIST with RLIF, it improves the accuracy to $98.54 \pm 0.03\%$, compared with $98.33 \pm 0.04\%$ for BPTT and $98.25 \pm 0.03\%$ for pp-prop. On SHD, COLA reaches $94.07 \pm 0.45\%$ with RLIF and $95.37 \pm 0.13\%$ with RadLIF, slightly exceeding the corresponding BPTT and pp-prop results in both settings. On DVS-Gesture with EGRU, COLA remains close to the offline baseline at $97.19 \pm 0.32\%$ versus $97.45 \pm 0.27\%$ for BPTT. The main exception is N-MNIST with RadLIF, where COLA reaches $93.56 \pm 0.12\%$ and trails both BPTT and pp-prop. Overall, these results show that the scalar closure transfers beyond tanh RNNs and remains effective on recurrent spiking cells, while some cell-task combinations remain challenging.

### 4.4. ConvRNN tasks

To test whether the same scalarization principle survives in higher-dimensional recurrent feature maps, we evaluate ConvRNNs on static-image refinement (Fashion-MNIST, Per-

muted MNIST) and event-frame benchmarks (DVS-Gesture, DVS-CIFAR10). The resulting picture is again task dependent: BPTT or TBPTT-10 remain strongest on all four tasks, but COLA stays competitive with e-prop and consistently outperforms UORO in the main table. Compared with the vanilla-RNN and SNN settings, however, the empirical gains are less pronounced. One possible reason is that COLA is most naturally aligned with settings where recurrent credit must track genuinely evolving temporal signals, whereas in ConvRNN refinement tasks, especially Fashion-MNIST and Permuted MNIST, the same input image is presented at every step and the recurrence mainly serves iterative spatial refinement rather than integrating new temporal evidence. Even so, the results suggest that per-channel loop-gain statistics are already informative in convolutional recurrence, although exact long-range gradients still matter for the strongest final accuracies. Appendix J gives the full ConvRNN tables and setup details.

### 4.5. LSTM scope probe

We also ran an empirical LSTM pilot on the same seven sequence tasks used in the vanilla-RNN suite: Adding, Lorenz image prediction, Row-MNIST, Row-CIFAR10, UCI HAR, PTB-Char, and WikiText-2 Char. LSTM couples a hidden state and a cell state, so we treat this experiment as an empirical scope probe. Its role is to test whether the same initialization-and-local-update recipe remains numerically serviceable once recurrence is no longer concentrated in a single chain; the appendix L gives the full comparison table.

### 4.6. Criticality Analysis

**Direct near-critical initialization.** Near-criticality is used only as a one-shot initializer, not as a training-long constraint. Appendix H shows that the direct estimators place the selected pre-training exponent consistently on the stable side; for the two-point estimator, $\lambda_{\mathrm{pre}}$ falls in a narrow band from about $-0.053$ to $-0.013$ across the 11 benchmarks, and the Lyapunov curves remain close to linear in $\log g$ near the boundary ($R^2 \approx 0.93$–$0.99$). Empirically, this is already accurate enough to recover scan-level performance on most tasks: for example, the two-point initializer gives Lorenz MSE 0.0057 versus 0.0058 for the scan and Permuted MNIST accuracy 0.9230 versus 0.9136. The one-time cost is small: across the 11 formal tasks, the median overhead is $0.163\%$ for the unit estimator and $0.317\%$ for the two-point estimator. Full tables are deferred to Appendices H and L (the apparent sweep-crossing range near 0.6–1.0 and the direct normalized gains near 2 differ only because the former uses the raw multiplier $\tilde{g}$ while the latter uses the normalized gain $g$).

**Hypothesis validation diagnostics.** COLA relies on two properties of the exact BPTT teaching signal: long-range

temporal correlation and long-range spatial correlation. Appendix G evaluates these properties in a controlled probe and on trained endpoints. In the rank-controlled probe, the exact teaching signal is almost perfectly supported by the two diagnostics (AR(1) $R^2$=$0.9944 \pm 0.0094$, $EVR_1$=$0.9999 \pm 0.0001$, effective rank $= 1.0001 \pm 0.0002$), while the IID control is also reasonably supportive, with AR(1) $R^2$=$0.8089 \pm 0.0334$, $EVR_1$=$0.9836 \pm 0.0112$, and effective rank $= 1.0339 \pm 0.0237$. After training, these signatures become task dependent. For example, Adding retains a stable band with temporal AR(1) $R^2$ between 0.5783 and 0.8097 and $EVR_1$ between 0.86 and 0.97, while UCI HAR retains $R^2$ between 0.7079 and 0.8270 and $EVR_1$ between 0.69 and 0.77. Lorenz, by contrast, remains temporally predictable while becoming more spatially diffuse. Detailed stable-band ranges appear in Appendix G.

## 5. Conclusion

This paper presented COLA, an online learning rule for RNNs that replaces temporal backpropagation with a closed-form, scalarized approximation of BPTT gradients. By operating in a near-critical dynamical regime and tracking only two scalars per unit, COLA achieves constant activation memory while preserving the outer-product update structure of BPTT. Experiments across regression, classification, and language modeling tasks show that COLA remains competitive with online baselines, is especially strong on long-horizon stability, and extends naturally to ConvRNNs and SNNs. Overall, COLA provides a physically grounded and memory-efficient alternative for online recurrent learning.

## 6. Limitations

First, the effectiveness of COLA's scalar closure depends on two assumptions about the teaching signal: long-range temporal correlation and long-range spatial correlation. Second, the method still inherits an intrinsic trade-off between online efficiency and gradient fidelity; in particular, on datasets with weaker temporal structure, such as ConvRNN static-image refinement where the same input is repeated at each step, COLA can remain slightly below BPTT. Third, the current implementation relies on near-critical initialization, although the measured overhead is small. Extending the framework to more modern architectures, such as ResNets (He et al., 2016) and Transformers (Vaswani et al., 2017), is a natural direction for future work.

## Acknowledgement

This work was supported in part by the Beijing Major Science and Technology Project under Contract no. Z251100008125055. This work was supported by Beijing Academy of Artificial Intelligence (BAAI). This work was

also supported by the National Natural Science Foundation of China (NSFC) under Grant No. 62576011.

## Impact Statement

This work advances methods for training recurrent neural networks with online, memory-efficient updates. No direct negative societal impacts are anticipated beyond those generally associated with general-purpose machine learning research.

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

## Appendix Roadmap

This supplement is organized to make it easy to locate experimental details, theory derivations, and diagnostic evidence:

- **Appendix A:** Derivation of the standard BPTT teaching-signal recursion and notation.

- **Appendix B:** From the BPTT recursion to a same-step implicit system and the scalar geometric-series closed form used by COLA.

- **Appendix C:** Role and interpretation of the temporal self-consistency criterion used to estimate $\mu_i$.

- **Appendix D:** Consistency and online estimation of the AR(1) coefficient $\alpha_i$.

- **Appendix E and Appendix F:** Closed-form and least-squares estimators for $\mu_i$ derived from self-consistency.

- **Appendix G:** Controlled probes, metric definitions, and task-level diagnostics validating the temporal and spatial assumptions.

- **Appendix H:** Finite-time Lyapunov estimation, gain normalization, direct near-critical initialization, and spatial mismatch robustness.

- **Appendix I:** Task descriptions and default hyperparameters for the main RNN and ConvRNN experiments.

- **Appendix J:** ConvRNN extension details and refinement benchmark results.

- **Appendix K:** SNN extension details and spiking recurrent benchmark results.

- **Appendix L:** Learning curves and supplementary experiments, including approximate-RTRL comparisons, EMA sensitivity, initialization overhead, tuned BPTT fairness checks, fast-target stress tests, and architectural extension pilots.

## A. Derivation of the BPTT Error Signal

**Purpose.** This appendix derives the standard BPTT recursion for the pre-activation teaching signal $\delta_t^{(i)}$ and fixes notation for subsequent appendices.

Define the pre-activation error (teaching signal)

$$\delta_t^{(i)} \triangleq \frac{\partial \mathcal{L}}{\partial x_t^{(i)}}. \tag{34}$$

STEP 1: SEPARATE THE ACTIVATION NONLINEARITY

Since $h_t^{(i)} = \phi(x_t^{(i)})$, the chain rule gives

$$\delta_t^{(i)} = \frac{\partial \mathcal{L}}{\partial h_t^{(i)}} \frac{\partial h_t^{(i)}}{\partial x_t^{(i)}}. \tag{35}$$

It remains to compute $\partial \mathcal{L} / \partial h_t^{(i)}$.

STEP 2: DECOMPOSE THE INFLUENCE OF $h_t^{(i)}$ ON THE TOTAL LOSS

Write the total loss as $\mathcal{L} = \sum_{k=1}^T \mathcal{L}_k$. The hidden state affects $\mathcal{L}$ through (i) the instantaneous output loss at time $t$ and (ii) its influence on future states. Therefore,

$$\frac{\partial \mathcal{L}}{\partial h_t^{(i)}} = \frac{\partial \mathcal{L}_t}{\partial h_t^{(i)}} + \frac{\partial \mathcal{L}_{>t}}{\partial h_t^{(i)}}, \qquad \mathcal{L}_{>t} \triangleq \sum_{k=t+1}^T \mathcal{L}_k. \tag{36}$$

STEP 3: CONTRIBUTION OF FUTURE LOSSES

By the recurrence $x_{t+1}^{(m)} = \sum_k W_{hh}^{mk} h_t^{(k)} + \cdots$, it follows that

$$\frac{\partial \mathcal{L}_{>t}}{\partial h_t^{(i)}} = \sum_{m=1}^{H} \frac{\partial \mathcal{L}}{\partial x_{t+1}^{(m)}} \frac{\partial x_{t+1}^{(m)}}{\partial h_t^{(i)}} = \sum_{m=1}^{H} \delta_{t+1}^{(m)} W_{hh}^{mi}. \tag{37}$$

STEP 4: COMBINE TERMS

Substituting Eq. (37) into Eq. (36) and then into Eq. (35) yields

$$\delta_t^{(i)} = \frac{\partial h_t^{(i)}}{\partial x_t^{(i)}} \left( \frac{\partial \mathcal{L}_t}{\partial h_t^{(i)}} + \sum_{m=1}^{H} W_{hh}^{mi} \delta_{t+1}^{(m)} \right), \tag{38}$$

which is equivalent to the vector recursion in Eq. (14).

## B. From the BPTT implicit system to a geometric-series closed form

This appendix provides a constructive derivation of the closed-form local teaching signal in Eq. (22). The argument proceeds in three stages:

1. Rewrite the BPTT recursion in vector form and absorb the cross-step dependency $\boldsymbol{\delta}_{t+1}$ into a same-step linear operator $\mathbf{A}_t$ via a short-horizon local linearization, yielding an implicit system $(\mathbf{I}_H - \mathbf{A}_t)\boldsymbol{\delta}_t = \partial \mathcal{L}_t / \partial \mathbf{x}_t$.

2. Under the structural hypotheses of spatially correlated (approximately rank-1) teaching signals and a dominant one-dimensional mode, close the same-step system elementwise via a per-unit loop gain, yielding the per-unit scalar geometric-series form in Eq. (22) (see Appendix G for the controlled-probe evidence that the dominant spatial rank is near one).

3. (Optional interpretation) In the near-critical yet stable regime, interpret the formal matrix solution $\boldsymbol{\delta}_t \approx (\mathbf{I}_H - \mathbf{A}_t)^{-1}(\partial \mathcal{L}_t / \partial \mathbf{x}_t)$ via the Neumann/path expansion, which makes explicit the notion of within-step recurrent "echoes".

**Why expand the Neumann series?** In the main text, once Assumption 2 closes the same-step system elementwise, the closed form in Eq. (22) follows directly and no matrix inverse is ever computed in the algorithm. The Neumann/path expansion is included here only to interpret the resolvent $(\mathbf{I}_H - \mathbf{A}_t)^{-1}$ as a sum of within-step multi-hop echoes on the recurrent graph, making explicit why near-critical but stable dynamics can induce long effective interaction ranges and long-range spatial correlation. Accordingly, we present the core scalar-closure derivation first and defer the Neumann/path interpretation to the end of this appendix.

### B.1. Vector form of the BPTT recursion

The scalar form of Eq. (14) is

$$\delta_t^{(i)} = \left( \frac{\partial \mathcal{L}_t}{\partial h_t^{(i)}} + \sum_m W_{hh}^{mi} \delta_{t+1}^{(m)} \right) \frac{\partial h_t^{(i)}}{\partial x_t^{(i)}}.$$

Stack all units into vectors as

$$\boldsymbol{\delta}_t = (\delta_t^{(1)}, \ldots, \delta_t^{(H)})^\top, \quad \frac{\partial \mathcal{L}_t}{\partial \mathbf{h}_t} = \left( \frac{\partial \mathcal{L}_t}{\partial h_t^{(1)}}, \ldots, \frac{\partial \mathcal{L}_t}{\partial h_t^{(H)}} \right)^\top, \quad \frac{\partial \mathcal{L}_t}{\partial \mathbf{x}_t} = \left( \frac{\partial \mathcal{L}_t}{\partial x_t^{(1)}}, \ldots, \frac{\partial \mathcal{L}_t}{\partial x_t^{(H)}} \right)^\top,$$

and denote

$$\mathbf{U}_t \equiv \mathrm{diag}\big(\phi'(x_t^{(1)}), \ldots, \phi'(x_t^{(H)})\big).$$

Then the recursion can be written in vector form as

$$\boldsymbol{\delta}_t = \frac{\partial \mathcal{L}_t}{\partial \mathbf{x}_t} + \mathbf{U}_t \mathbf{W}_{hh}^\top \boldsymbol{\delta}_{t+1}. \tag{39}$$

Thus, BPTT induces a linear but cross-time recursion in $\boldsymbol{\delta}_t$.

## B.2. From cross-time recursion to a same-step implicit system

We seek an approximation that depends only on quantities at the current time step. Accordingly, the cross-step dependence $\boldsymbol{\delta}_{t+1}$ is approximated as a linear function of $\boldsymbol{\delta}_t$.

Under Assumption 1 (long-range temporal correlations / short-window continuation), the cross-step dependence is approximated by a per-unit linear map on the true BPTT pre-activation gradient:

$$\delta_{t+1}^{(i)} \approx \alpha_i \delta_t^{(i)}.$$

This is an empirical short-window model of $\boldsymbol{\delta}_t$ that is directly validated in Appendix G (relative MSE 0.0044 with intercept; 0.0061 without intercept). Let

$$\mathbf{D}_\alpha \equiv \operatorname{diag}(\alpha_1, \ldots, \alpha_H),$$

so that, in vector form,

$$\boldsymbol{\delta}_{t+1} \approx \mathbf{D}_\alpha \boldsymbol{\delta}_t.$$

Substituting this approximation into the BPTT vector recursion in Eq. (39) yields

$$\boldsymbol{\delta}_t \approx \frac{\partial \mathcal{L}_t}{\partial \mathbf{x}_t} + \mathbf{U}_t \mathbf{W}_{hh}^\top \mathbf{D}_\alpha \boldsymbol{\delta}_t.$$

Rearranging terms gives

$$\left(\mathbf{I}_H - \mathbf{U}_t \mathbf{W}_{hh}^\top \mathbf{D}_\alpha\right) \boldsymbol{\delta}_t \approx \frac{\partial \mathcal{L}_t}{\partial \mathbf{x}_t}.$$

Define

$$\mathbf{A}_t \equiv \mathbf{U}_t \mathbf{W}_{hh}^\top \mathbf{D}_\alpha, \qquad \text{i.e.} \quad (\mathbf{A}_t)_{ij} = \frac{\partial h_t^{(i)}}{\partial x_t^{(i)}} W_{hh}^{ji} \alpha_j, \tag{40}$$

so the above equation becomes

$$(\mathbf{I}_H - \mathbf{A}_t) \boldsymbol{\delta}_t \approx \frac{\partial \mathcal{L}_t}{\partial \mathbf{x}_t}. \tag{41}$$

This step does not introduce any additional structural assumptions: the structure of $\mathbf{A}_t$ is fully determined by

$$\mathbf{U}_t, \ \mathbf{W}_{hh}, \ \alpha_i$$

which are defined in the main text, together with the empirical continuation model $\boldsymbol{\delta}_{t+1} \approx \mathbf{D}_\alpha \boldsymbol{\delta}_t$.

**Spectral radius and stability** For any submultiplicative matrix norm $\|\cdot\|$, we have

$$\rho(\mathbf{A}_t) \leq \|\mathbf{A}_t\| \leq \|\mathbf{U}_t\| \|\mathbf{W}_{hh}^\top\| \|\mathbf{D}_\alpha\|.$$

In particular, with the spectral norm $\|\cdot\|_2$,

$$\rho(\mathbf{A}_t) \leq \|\mathbf{A}_t\|_2 \leq \left(\max_i |\frac{\partial h_t^{(i)}}{\partial x_t^{(i)}}|\right) \|\mathbf{W}_{hh}\|_2 \left(\max_i |\alpha_i|\right).$$

Assumption 1 requires a near-critical yet stable operating point, namely that the Jacobian

$$\mathbf{J}_t = \operatorname{diag}\left(\frac{\partial \mathbf{h}_t}{\partial \mathbf{x}_t}\right) \mathbf{W}_{hh}$$

has spectral radius $\rho(\mathbf{J}_t) < 1$ and that $\max_i |\alpha_i| < 1$. This motivates operating in a regime where $\rho(\mathbf{A}_t)$ is close to but typically below 1, so that $(\mathbf{I}_H - \mathbf{A}_t)^{-1}$ is well-defined (and admits a Neumann expansion; see Sec. B.4). Our gain sweep explicitly targets the stable edge-of-chaos transition ($\lambda_{\max} \approx 0^-$), placing $\rho(\mathbf{J}_t)$ close to one; empirically $\rho(\mathbf{A}_t)$ is likewise near-critical. In practice, $|\mu_i \frac{\partial h_t^{(i)}}{\partial x_t^{(i)}}| < 1$ is enforced by projection (Eq. (31)), guaranteeing convergence of the scalar series used in the final closed form. In the archived rank-1 controlled probe, the trajectory-averaged spectral radius of $\mathbf{A}_t$ is $0.916 \pm 0.098$ across seeds, with per-seed maxima averaging $1.030 \pm 0.013$. The projection keeps $\max_i |\mu_i \frac{\partial h_t^{(i)}}{\partial x_t^{(i)}}|$ below 1 (near the clip cap).

### B.3. Scalarization via a per-unit loop-gain closure

The same-step implicit system (Eq. (41)) can be written as

$$\boldsymbol{\delta}_t \approx \frac{\partial \mathcal{L}_t}{\partial \mathbf{x}_t} + \mathbf{A}_t \boldsymbol{\delta}_t, \qquad \mathbf{A}_t \equiv \mathbf{U}_t \mathbf{W}_{hh}^\top \mathbf{D}_\alpha.$$

For a fixed unit $i$, the $i$th component reads

$$\delta_t^{(i)} \approx \frac{\partial \mathcal{L}_t}{\partial x_t^{(i)}} + \frac{\partial h_t^{(i)}}{\partial x_t^{(i)}} \sum_{j=1}^{H} W_{hh}^{ji} \, \alpha_j \, \delta_t^{(j)}. \tag{42}$$

**From long-range spatial correlation to a scalar loop gain** Assumption 2 states that the within-step teaching signal exhibits long-range spatial correlation: for a fixed reference unit $i$, each component $\delta_t^{(j)}$ is approximately proportional to $\delta_t^{(i)}$,

$$\delta_t^{(j)} \approx \beta_{ji}(t) \, \delta_t^{(i)}, \qquad \beta_{ji}(t) \text{ slowly varying in } t. \tag{43}$$

Applying Eq. (43) directly to the recurrent summation in Eq. (42) gives, step-by-step,

$$\sum_{j=1}^{H} W_{hh}^{ji} \, \alpha_j \, \delta_t^{(j)} \approx \sum_{j=1}^{H} W_{hh}^{ji} \, \alpha_j \left( \beta_{ji}(t) \, \delta_t^{(i)} \right)$$

$$= \left( \sum_{j=1}^{H} W_{hh}^{ji} \, \alpha_j \, \beta_{ji}(t) \right) \delta_t^{(i)}. \tag{44}$$

Therefore the recurrent summation is (approximately) a scalar multiple of the local teaching signal. Define the induced per-unit loop gain by

$$\mu_i \approx \sum_{j=1}^{H} W_{hh}^{ji} \, \alpha_j \, \beta_{ji}(t).$$

Since both $\beta_{ji}(t)$ (spatial-correlation factor) and $\alpha_j$ (temporal-persistence factor) vary slowly, this coefficient is expected to vary slowly as well, motivating COLA's online tracking of a per-unit loop gain.

**Closed form induced by the loop gain** Substituting Eq. (44) into Eq. (42) yields

$$\delta_t^{(i)} \approx \frac{\partial \mathcal{L}_t / \partial x_t^{(i)}}{1 - \mu_i \frac{\partial h_t^{(i)}}{\partial x_t^{(i)}}},$$

which is the same scalar closure used in the main text.

**COLA approximation** COLA tracks this loop gain online, giving the closed form

$$\widetilde{\delta}_t^{(i)} \approx \frac{\partial \mathcal{L}_t / \partial x_t^{(i)}}{1 - \mu_i \frac{\partial h_t^{(i)}}{\partial x_t^{(i)}}}, \tag{45}$$

which matches Eq. (22) in the main text. Appendix G assesses the quality of this scalar closure empirically through controlled-probe diagnostics and task-level diagnostics for the temporal and spatial assumptions used to estimate $\mu_i$.

This completes the derivation from the BPTT recursion to the scalar closed form. The argument relies only on:

- Assumption 1 (short-window continuation of the teaching signal), used to linearize $\boldsymbol{\delta}_{t+1} \approx \mathbf{D}_\alpha \boldsymbol{\delta}_t$.

- A per-unit loop-gain closure, namely that the recurrent summation in Eq. (42) can be tracked by a slowly varying online estimate $\mu_i$.

- Numerical stability, enforced by bounding $|\mu_i \frac{\partial h_t^{(i)}}{\partial x_t^{(i)}}| < 1$ (Eq. (31)).

## B.4. Interpretation: resolvent as a Neumann/path expansion

This subsection is included for intuition (not for computation). Starting from the same-step implicit system (Eq. (41)), the formal solution has the form

$$\boldsymbol{\delta}_t \approx (\mathbf{I}_H - \mathbf{A}_t)^{-1} \frac{\partial \mathcal{L}_t}{\partial \mathbf{x}_t}.$$

Denote the resolvent (Green's function) by $\mathbf{G}_t \triangleq (\mathbf{I}_H - \mathbf{A}_t)^{-1}$. When $\rho(\mathbf{A}_t) < 1$, the inverse exists and admits the Neumann expansion:

$$\mathbf{G}_t = \sum_{k=0}^{\infty} (\mathbf{A}_t)^k. \tag{46}$$

Therefore, $\boldsymbol{\delta}_t$ decomposes into a superposition of within-step "echoes":

$$\boldsymbol{\delta}_t \approx \sum_{k=0}^{\infty} (\mathbf{A}_t)^k \frac{\partial \mathcal{L}_t}{\partial \mathbf{x}_t}.$$

In component form, the $k$-hop contribution is a sum over length-$k$ directed paths from $j$ to $i$,

$$\left[(\mathbf{A}_t)^k\right]_{ij} = \sum_{i_1, \ldots, i_{k-1}} (\mathbf{A}_t)_{i i_{k-1}} (\mathbf{A}_t)_{i_{k-1} i_{k-2}} \cdots (\mathbf{A}_t)_{i_1 j}, \tag{47}$$

This makes explicit how near-critical but stable dynamics (slow decay in $k$) can induce long effective interaction ranges and long-range spatial correlations across units.

# C. Derivation and interpretation of temporal self-consistency

This appendix clarifies the role of the temporal self-consistency relation

$$\widetilde{\delta}_t^{(i)} \approx \alpha_i \, \widetilde{\delta}_{t-1}^{(i)},$$

where $\widetilde{\delta}_t^{(i)}$ is the closed-form local teaching signal in Eq. (22). The key point is that this relation is not meant as an exact identity or theorem about the surrogate itself. Rather, it is the identification criterion that ties the unknown loop gain $\mu_i$ in Eq. (22) to the only temporal structure assumed for the target BPTT teaching signal.

**Step 1: Temporal structure of the target signal** Assumption 1 states that, over short windows in the stable near-critical regime, the true BPTT teaching signal is well described by a per-unit continuation law:

$$\delta_t^{(i)} \approx \alpha_i \delta_{t-1}^{(i)}.$$

Appendix G gives controlled-probe diagnostics supporting this short-window model and also shows that its fidelity is task dependent.

**Step 2: Why an extra criterion is needed for $\mu_i$** After the spatial closure, COLA models the teaching signal by

$$\widetilde{\delta}_t^{(i)}(\mu_i) = \frac{\partial \mathcal{L}_t / \partial x_t^{(i)}}{1 - \mu_i \frac{\partial h_t^{(i)}}{\partial x_t^{(i)}}}.$$

At this point $\mu_i$ is still unknown. The closed form itself does not determine $\mu_i$ from one time step alone, so an additional online identification criterion is required. The most natural choice under Assumption 1 is to require that the surrogate sequence obey the same short-window continuation law as the target sequence.

**Step 3: Identification by temporal self-consistency** We therefore choose $\mu_i$ so that

$$\widetilde{\delta}_t^{(i)}(\mu_i) \approx \alpha_i \widetilde{\delta}_{t-1}^{(i)}(\mu_i).$$

This is the temporal self-consistency condition used in the main text (Eq. (26)). It should be read as a moment-matching or model-consistency condition: if $\widetilde{\delta}_t^{(i)}$ is to approximate $\delta_t^{(i)}$, then it should reproduce the temporal pattern that Assumption 1 attributes to $\delta_t^{(i)}$. Substituting Eq. (22) into this condition yields the linear relation in Eq. (27), and Appendix E and Appendix F derive the corresponding one-step and exponentially weighted least-squares estimators for $\mu_i$.

**Interpretation** The temporal self-consistency relation is therefore an estimation principle, not an exact identity. When Assumption 1 is accurate and the scalar closure is good, the residual of this relation is small and the resulting $\mu_i$ estimate is informative. When the short-window continuation model weakens, the same estimator naturally becomes less reliable, which is why the paper presents Assumption 1 as an empirically supported but task-dependent approximation rather than a universal law.

## D. Consistency and implementation of the least-squares estimate for $\alpha_i$

**Purpose.** This appendix derives the least-squares estimator for the AR(1) continuation coefficient $\alpha_i$ and summarizes its online EMA implementation.

Consider the zero-intercept AR(1) model $\widetilde{\delta}_t^{(i)} \approx \alpha_i \widetilde{\delta}_{t-1}^{(i)}$. The least-squares objective is

$$\min_{\alpha_i} \mathbb{E}\left[\left(\widetilde{\delta}_t^{(i)} - \alpha_i \widetilde{\delta}_{t-1}^{(i)}\right)^2\right].$$

Differentiating with respect to $\alpha_i$ and setting the derivative to zero yields

$$\frac{\partial}{\partial \alpha_i} \mathbb{E}\left[\left(\widetilde{\delta}_t^{(i)} - \alpha_i \widetilde{\delta}_{t-1}^{(i)}\right)^2\right] = \mathbb{E}\left[-2\widetilde{\delta}_t^{(i)}\widetilde{\delta}_{t-1}^{(i)} + 2\alpha_i(\widetilde{\delta}_{t-1}^{(i)})^2\right] = 0.$$

Therefore,

$$-2\mathbb{E}\left[\widetilde{\delta}_t^{(i)}\widetilde{\delta}_{t-1}^{(i)}\right] + 2\alpha_i\mathbb{E}\left[(\widetilde{\delta}_{t-1}^{(i)})^2\right] = 0,$$

and the closed-form solution is

$$\alpha_i^\star = \frac{\mathbb{E}[\widetilde{\delta}_t^{(i)}\widetilde{\delta}_{t-1}^{(i)}]}{\mathbb{E}[(\widetilde{\delta}_{t-1}^{(i)})^2]}.$$

Under stationarity or ergodicity, time averages can replace expectations. Exponentially weighted averaging (EWA) is an efficient online surrogate for such time averages, yielding a consistent estimator. Finally, clipping $|\alpha_i| \leq \alpha_{\max} < 1$ enforces stability of the modeled dynamics, consistent with the near-critical regime assumed by COLA.

## E. Algebraic derivation of the one-step closed form for $\mu_i$

This appendix provides the algebraic steps that derive the one-step closed-form estimator in Eq. (49) from the self-consistency constraint. We start from

$$\frac{\partial \mathcal{L}_t/\partial x_t^{(i)}}{1 - \mu_i \frac{\partial h_t^{(i)}}{\partial x_t^{(i)}}} \approx \alpha_i \frac{\partial \mathcal{L}_{t-1}/\partial x_{t-1}^{(i)}}{1 - \mu_i \phi'(x_{t-1}^{(i)})}. \tag{48}$$

**Step 1: Cross-multiply**

$$\frac{\partial \mathcal{L}_t}{\partial x_t^{(i)}}\left(1 - \mu_i \phi'(x_{t-1}^{(i)})\right) \approx \alpha_i \frac{\partial \mathcal{L}_{t-1}}{\partial x_{t-1}^{(i)}}\left(1 - \mu_i \phi'(x_t^{(i)})\right).$$

**Step 2: Expand products**

$$\frac{\partial \mathcal{L}_t}{\partial x_t^{(i)}} - \mu_i \phi'(x_{t-1}^{(i)})\frac{\partial \mathcal{L}_t}{\partial x_t^{(i)}} \approx \alpha_i \frac{\partial \mathcal{L}_{t-1}}{\partial x_{t-1}^{(i)}} - \mu_i \alpha_i \phi'(x_t^{(i)})\frac{\partial \mathcal{L}_{t-1}}{\partial x_{t-1}^{(i)}}.$$

**Step 3: Collect $\mu_i$ terms**

$$\mu_i \alpha_i \phi'(x_t^{(i)})\frac{\partial \mathcal{L}_{t-1}}{\partial x_{t-1}^{(i)}} - \mu_i \phi'(x_{t-1}^{(i)})\frac{\partial \mathcal{L}_t}{\partial x_t^{(i)}} \approx \alpha_i \frac{\partial \mathcal{L}_{t-1}}{\partial x_{t-1}^{(i)}} - \frac{\partial \mathcal{L}_t}{\partial x_t^{(i)}}.$$

**Step 4: Factorize**

$$\mu_i\left(\alpha_i \phi'(x_t^{(i)})\frac{\partial \mathcal{L}_{t-1}}{\partial x_{t-1}^{(i)}} - \phi'(x_{t-1}^{(i)})\frac{\partial \mathcal{L}_t}{\partial x_t^{(i)}}\right) \approx \alpha_i \frac{\partial \mathcal{L}_{t-1}}{\partial x_{t-1}^{(i)}} - \frac{\partial \mathcal{L}_t}{\partial x_t^{(i)}}.$$

**Step 5: Solve for** $\mu_i$   If $\alpha_i \phi'(x_t^{(i)})\frac{\partial \mathcal{L}_{t-1}}{\partial x_{t-1}^{(i)}} - \phi'(x_{t-1}^{(i)})\frac{\partial \mathcal{L}_t}{\partial x_t^{(i)}} \neq 0$, dividing both sides yields

$$\mu_i^{(t)} = \frac{\alpha_i \frac{\partial \mathcal{L}_{t-1}}{\partial x_{t-1}^{(i)}} - \frac{\partial \mathcal{L}_t}{\partial x_t^{(i)}}}{\alpha_i \phi'(x_t^{(i)})\frac{\partial \mathcal{L}_{t-1}}{\partial x_{t-1}^{(i)}} - \phi'(x_{t-1}^{(i)})\frac{\partial \mathcal{L}_t}{\partial x_t^{(i)}}}, \tag{49}$$

which is the desired one-step closed form.

## F. Exponentially weighted least squares for $\mu_i$

**Purpose.**   This appendix derives the exponentially weighted least-squares update used to estimate the loop gain $\mu_i$ online from the linear relation in Eq. (27).

We aggregate the linear relations $\mu_i A_i^\tau \approx B_i^\tau$ across time using exponentially decaying weights $\omega_\tau = (1 - \rho_\mu)\rho_\mu^{t-\tau}$ and minimize the weighted squared error:

$$\min_{\mu_i} \mathcal{J}(\mu_i) = \min_{\mu_i} \sum_{\tau=1}^{t} \omega_\tau (\mu_i A_i^\tau - B_i^\tau)^2. \tag{50}$$

**Step 1: Objective**

$$\mathcal{J}(\mu_i) = \sum_{\tau=1}^{t} \omega_\tau (\mu_i A_i^\tau - B_i^\tau)^2. \tag{51}$$

**Step 2: Differentiate**

$$\frac{\partial \mathcal{J}}{\partial \mu_i} = \sum_{\tau=1}^{t} 2\omega_\tau (\mu_i A_i^\tau - B_i^\tau)A_i^\tau. \tag{52}$$

**Step 3: Set the gradient to zero**   The optimum satisfies

$$\sum_{\tau=1}^{t} \omega_\tau (\mu_i A_i^\tau - B_i^\tau)A_i^\tau = 0. \tag{53}$$

**Step 4: Collect $\mu_i$ terms**

$$\mu_i \sum_{\tau=1}^{t} \omega_\tau (A_i^\tau)^2 = \sum_{\tau=1}^{t} \omega_\tau A_i^\tau B_i^\tau. \tag{54}$$

**Step 5: Closed form**   If $\sum_\tau \omega_\tau (A_i^\tau)^2 \neq 0$, then

$$\mu_i = \frac{\sum_{\tau=1}^{t} \omega_\tau A_i^\tau B_i^\tau}{\sum_{\tau=1}^{t} \omega_\tau (A_i^\tau)^2}, \tag{55}$$

which matches Eq. (29) in the main text.

For an online implementation, the numerator and denominator can be updated via exponentially weighted averages:

$$S_{AB}(t) = \sum_{\tau=1}^{t} (1 - \rho_\mu)\rho_\mu^{t-\tau} A_i^\tau B_i^\tau$$
$$= \rho_\mu S_{AB}(t-1) + (1 - \rho_\mu)A_i^t B_i^t,$$
$$S_{A^2}(t) = \sum_{\tau=1}^{t} (1 - \rho_\mu)\rho_\mu^{t-\tau} (A_i^\tau)^2$$
$$= \rho_\mu S_{A^2}(t-1) + (1 - \rho_\mu)(A_i^t)^2,$$

so that

$$\mu_i(t) = \frac{S_{AB}(t)}{S_{A^2}(t) + \varepsilon_\mu}, \tag{56}$$

with a projection $\Pi$ applied in the main text to enforce $|\mu_i \frac{\partial h_t^{(i)}}{\partial x_t^{(i)}}|$ bounds for numerical stability.

## G. Hypothesis Validation

This appendix separates two questions. First, does the scalar closure become accurate in a controlled near-critical probe where the recurrent dynamics are easy to interpret? Second, after training on real tasks, do the same temporal and spatial signatures remain visible in a stable gain band? Throughout this appendix, all diagnostics are computed from the exact total BPTT pre-activation teaching signal $\boldsymbol{\delta}_t^{\text{true}}$.

### G.1. Probe setup and noise model

The controlled probe fixes recurrent weights and compares COLA's closed-form signal $\widehat{\boldsymbol{\delta}}_t$ against the exact BPTT teaching signal $\boldsymbol{\delta}_t^{\text{true}}$ (Appendix A). Inputs are i.i.d. Gaussian,

$$\mathbf{u}_t \sim \mathcal{N}(0, \sigma_{\text{in}}^2),$$

or alternatively Poisson with mean removed, $\mathbf{u}_t \sim \text{Poisson}(\lambda) - \lambda$. Targets follow a stationary AR(1) process,

$$\mathbf{y}_t = \rho_y \mathbf{y}_{t-1} + \boldsymbol{\varepsilon}_t, \qquad \boldsymbol{\varepsilon}_t \sim \mathcal{N}\left(0, \sigma_y^2(1 - \rho_y^2)\mathbf{I}\right),$$

so that $\text{Var}(\mathbf{y}_t) = \sigma_y^2$. The instantaneous output error may be low-pass filtered by

$$\tilde{\mathbf{e}}_t = \rho_e \tilde{\mathbf{e}}_{t-1} + (1 - \rho_e) \, \partial \mathcal{L}_t / \partial \mathbf{o}_t.$$

Unless stated otherwise, we use $H{=}128$, $T{=}700$, burn-in $T_0{=}100$, batch size 16, and 21 random seeds.

### G.2. Controlled near-critical probe

The rank-controlled probe family is

$$W_{\text{base}} = \sqrt{1 - \text{frac}}\,\frac{G}{\sqrt{H}} \; + \; \sqrt{\text{frac}}\,\frac{1}{RH}VV^\top, \qquad W_{hh} = g_{\text{crit}}\, W_{\text{base}},$$

where $g_{\text{crit}}$ is chosen per seed so that the maximum Lyapunov exponent is close to $0^-$. For IID probes we set $\text{frac} = 0$; for low-rank probes we align the readout with the low-rank basis so that the closure mechanism can be inspected directly. For the rank-1 probe, $g_{\text{crit}} \in [0.874, 1.196]$ and $|\lambda_{\max}| \leq 4.3 \times 10^{-3}$.

In the controlled probe, $g_{\text{crit}}$ stays near 1 because the base operator is fixed and already normalized. That statement should not be extrapolated to task-facing training: Appendix H shows that once the same normalization is applied to actual trainable models, the task-aware direct estimates are typically around 2 and can be much larger on row-sequential vision tasks. The probe therefore supports the mechanism, not a universal gain constant.

**Why the rank-controlled probe is still useful.** The low-rank probe does not claim that trained recurrent weights are literally low rank. Its role is narrower: it isolates the one-dimensional closure mechanism in a setting where the dominant spatial mode is known by construction. Transfer to full-rank training is therefore treated as an empirical question rather than an assumption. For that reason, we also report IID probes and task-level endpoint scans below.

### G.3. Definitions of the diagnostics

We report three quantities that directly track Assumptions 1–2.

**Temporal AR(1) fit.** Let $\bar{\boldsymbol{\delta}}_t^{\text{true}} \in \mathbb{R}^H$ denote the batch-averaged true teaching signal after burn-in, and let $W$ be the short fit window. For each eligible time $t$ and each unit $i$, we fit the affine one-step model

$$\bar{\delta}_\tau^{\text{true},(i)} \approx a_i^{(t)} \, \bar{\delta}_{\tau-1}^{\text{true},(i)} + b_i^{(t)}, \qquad \tau = t - W + 1, \ldots, t,$$

on the preceding window and then use the fitted pair $(a_i^{(t)}, b_i^{(t)})$ to make the one-step prediction

$$\widehat{\delta}_t^{(i)} = a_i^{(t)} \, \bar{\delta}_{t-1}^{\text{true},(i)} + b_i^{(t)}.$$

The temporal score pools these sliding-window one-step predictions over all valid $t$ and all units:

$$R^2 = 1 - \frac{\sum_{i,t} \left( \bar{\delta}_t^{\text{true},(i)} - \widehat{\delta}_t^{(i)} \right)^2}{\sum_{i,t} \left( \bar{\delta}_t^{\text{true},(i)} - \bar{\delta}_{\text{win}}^{(i)} \right)^2},$$

where $\bar{\delta}_{\text{win}}^{(i)}$ is the mean of the collected target values for unit $i$. Thus $R^2 \approx 1$ means that a one-step AR(1)+bias model explains most of the post-burn-in temporal variation.

**Spatial concentration.** Let $X \in \mathbb{R}^{T' \times H}$ be the time-by-unit matrix obtained by stacking the post-burn-in vectors $\bar{\delta}_t^{\text{true}}$ row by row over time. If $X = U\Sigma V^\top$ has singular values $\{\sigma_k\}$, then the effective rank (participation ratio) is

$$r_{\text{eff}}(X) \triangleq \frac{\left( \sum_k \sigma_k^2 \right)^2}{\sum_k \sigma_k^4},$$

and the leading-mode energy fraction is

$$EVR_1(X) \triangleq \frac{\sigma_1^2}{\sum_k \sigma_k^2}.$$

The quantity $r_{\text{eff}}$ measures how many singular directions carry non-negligible energy, while $EVR_1$ measures how much of the total energy is captured by the top mode alone. Therefore, $r_{\text{eff}} \approx 1$ together with $EVR_1 \approx 1$ indicates that the post-burn-in teaching-signal trajectory is strongly concentrated in a single dominant spatial direction.

*Table 4.* Compact controlled-probe diagnostics on the exact BPTT teaching signal in near-critical low-rank and IID settings. Values are mean $\pm$ std across seeds.

| Quantity | Low-rank | IID |
|---|---|---|
| Temporal AR(1)+bias one-step $R^2$ | $0.9944 \pm 0.0094$ | $0.8089 \pm 0.0334$ |
| Teaching-signal $EVR_1$ | $0.9999 \pm 0.0001$ | $0.9836 \pm 0.0112$ |
| Teaching-signal effective rank | $1.0001 \pm 0.0002$ | $1.0339 \pm 0.0237$ |

Table 4 shows that the rank-controlled probe provides nearly ideal support for both assumptions, while the IID probe remains strongly supportive but weaker. This is the intended interpretation of the mechanism: the closure is most accurate in a near-critical low-dimensional regime, yet substantial temporal persistence and spatial concentration survive even without explicit low-rank structure.

### G.4. Task-level stable gain-band diagnostics

To complement the controlled probe, we rescan trained task endpoints by multiplicatively perturbing the recurrent gain around each task-specific reference operator and recomputing the same diagnostics on $\delta_t^{\text{true}}$. The gain band is therefore a relative multiplier around that trained endpoint, not the absolute normalized gain used by the direct initializer below. We retain contiguous gain intervals for which the dynamics remain stable ($\lambda_{\max} \leq 0$). Table 5 lists the resulting metric ranges.

*Table 5.* Task-level stable gain bands for the exact total BPTT teaching signal on trained endpoints. Each metric gives the range observed within the corresponding stable gain band.

| Task | Gain band | $\lambda_{\max}$ band | Temporal AR(1) $R^2$ | $EVR_1$ | Effective rank |
|---|---|---|---|---|---|
| Adding Task | 0.6–1.0 | $-0.3307$ to $-0.0231$ | 0.5783–0.8097 | 0.86–0.97 | 1.05–1.32 |
| Lorenz Image | 0.6–1.4 | $-1.0657$ to $-0.6956$ | 0.6248–0.8247 | 0.56–0.72 | 1.92–2.39 |
| Row-MNIST | 0.6–1.0 | $-0.1967$ to $-0.0447$ | 0.5408–0.7654 | 0.73–0.88 | 1.29–1.81 |
| Row-CIFAR10 | 0.6–1.0 | $-0.3707$ to $-0.0047$ | 0.5890–0.5940 | 0.72–0.73 | 1.33–1.79 |
| UCI HAR | 0.6–1.4 | $-0.3480$ to $-0.1027$ | 0.7079–0.8270 | 0.69–0.77 | 1.74–2.35 |

All five tasks retain nontrivial stable bands rather than isolated supportive points. Adding and UCI HAR show the clearest joint temporal and spatial support; Row-MNIST and Row-CIFAR10 remain supportive but over narrower temporal ranges; and Lorenz retains strong temporal persistence while exhibiting weaker spatial concentration. We omit the character-level language-model tasks from this compact table because the present stable-band scan does not yield a comparably clean supportive interval for them.

## H. Lyapunov diagnostics, direct Initialization, and spatial mismatch

### H.1. Finite-time Lyapunov estimator and normalized coordinate

For a recurrent model driven by data, we estimate finite-time Lyapunov exponents along a trajectory using the standard QR method (Benettin et al., 1980; Wolf et al., 1985). Starting from an orthonormal basis $\mathbf{Q}_0$, we iterate $\mathbf{Z}_t = \mathbf{J}_t \mathbf{Q}_{t-1}$ and compute $\mathbf{Z}_t = \mathbf{Q}_t \mathbf{R}_t$. Averaging the logarithms of the diagonal of $\mathbf{R}_t$ yields the Lyapunov spectrum, and we use the maximum exponent $\lambda_{\max}$ as the stability diagnostic throughout the paper.

Direct gain estimation is performed in the normalized coordinate introduced in Sec. 3.2, i.e. after factoring $\mathbf{W}_{hh} = g \overline{\mathbf{W}}_{hh}$ with $\|\overline{\mathbf{W}}_{hh}\|_{\mathrm{op}} = 1$. This removes arbitrary scale variation from the random initializer and makes gains comparable across tasks.

### H.2. Direct estimator derivation and why gain $1$ is not enough

The finite-time estimator used in Sec. 3.2 can be written in the normalized coordinate as a product of driven Jacobians

$$\mathbf{J}_t(g) = \mathbf{D}_t(g)\, g\, \overline{\mathbf{W}}_{hh}, \qquad \mathbf{D}_t(g) \triangleq \mathrm{diag}\!\left(\frac{\partial \mathbf{h}_t}{\partial \mathbf{x}_t}\right), \tag{57}$$

where the same decomposition applies to ConvRNNs and the effective recurrent chains of the SNN cells discussed in Appendix K. Along the QR trajectory,

$$\log \|\mathbf{J}_t(g)\mathbf{q}_{t-1}\| = \log g + \log \|\mathbf{D}_t(g)\overline{\mathbf{W}}_{hh}\mathbf{q}_{t-1}\|. \tag{58}$$

If the second term varies slowly inside the local stable-side band, then the dependence of the maximum Lyapunov exponent on $\log g$ is locally close to affine:

$$\lambda_{\max}(g) \approx a + b \log g. \tag{59}$$

Equivalently, if we introduce the log-gain coordinate

$$x \triangleq \log g, \qquad \phi(x) \triangleq \lambda_{\max}(e^x),$$

then the estimator assumes that $\phi(x)$ is locally well approximated by a first-order model around the reference point $x = 0$ (i.e. $g = 1$). A Taylor expansion gives

$$\phi(x) = \phi(0) + \phi'(0)x + \frac{1}{2}\phi''(\xi_x)x^2 \tag{60}$$

for some $\xi_x$ between $0$ and $x$. The direct initializer is therefore accurate when the local curvature $|\phi''|$ is small over the stable-side interval used for probing.

Let $\lambda_1 \triangleq \lambda_{\max}(1)$. Because $\log 1 = 0$, the affine model gives $a \approx \lambda_1$. The critical gain is defined by $\lambda_{\max}(g^\star) \approx 0$, hence

$$0 \approx \lambda_1 + b \log g^\star \qquad \Longrightarrow \qquad g^\star \approx \exp\!\left(-\frac{\lambda_1}{b}\right). \tag{61}$$

To see what the unit estimator assumes, write

$$\phi(x) = x + c(x), \qquad c(x) \triangleq \lambda_{\max}(e^x) - x.$$

The explicit scalar gain contributes the $x$ term, so $\phi'(0) = 1 + c'(0)$. The unit estimator freezes the residual contraction term locally, i.e. assumes $c'(0) \approx 0$, or equivalently $b \approx 1$. This yields

$$g_{\mathrm{unit}} = \exp(-\lambda_1). \tag{62}$$

*Table 6.* Direct near-critical initialization in normalized coordinates. $\lambda_{\mathrm{pre}}(g)$ denotes the measured pre-training Lyapunov exponent after applying the selected direct initializer at gain $g$. Columns "Unit", "Two-point", and "Scan" show matched representative task-facing reruns from the archived experiment bundle used to compare direct initialization against exhaustive gain sweeps.

| Task | Metric | $g_{\mathrm{unit}}$ | $\lambda_{\mathrm{pre}}(g_{\mathrm{unit}})$ | Unit | $g_{\mathrm{two}}$ | $\lambda_{\mathrm{pre}}(g_{\mathrm{two}})$ | Two-point | Scan |
|------|--------|------|------|------|------|------|------|------|
| Adding | MSE $\downarrow$ | 1.9014 | $-0.0832$ | 0.0061 | 2.0920 | $-0.0479$ | 0.0077 | 0.0029 |
| Lorenz | MSE $\downarrow$ | 1.9690 | $-0.0581$ | 0.0051 | 2.0981 | $-0.0194$ | 0.0057 | 0.0058 |
| PTB | Loss $\downarrow$ | 1.9261 | $-0.0424$ | 1.8294 | 2.0150 | $-0.0205$ | 1.8316 | 1.8416 |
| WikiText-2 | Loss $\downarrow$ | 1.8529 | $-0.0344$ | 2.4385 | 1.9220 | $-0.0528$ | 2.4362 | 2.4136 |
| Row-MNIST | Acc $\uparrow$ | 2.3189 | $-0.1060$ | 0.9508 | 2.6180 | $-0.0373$ | 0.9475 | 0.9543 |
| Row-CIFAR10 | Acc $\uparrow$ | 2.7674 | $-0.1559$ | 0.4025 | 3.3270 | $-0.0150$ | 0.3941 | 0.4513 |
| UCI HAR | Acc $\uparrow$ | 1.9906 | $-0.0634$ | 0.8680 | 2.1346 | $-0.0401$ | 0.8531 | 0.8622 |
| Fashion-MNIST | Acc $\uparrow$ | 2.0544 | $-0.0511$ | 0.8369 | 2.1707 | $-0.0140$ | 0.8315 | 0.8452 |
| Permuted MNIST | Acc $\uparrow$ | 2.0525 | $-0.0555$ | 0.9158 | 2.1800 | $-0.0177$ | 0.9230 | 0.9136 |
| DVS-Gesture | Acc $\uparrow$ | 2.0239 | $-0.0426$ | 0.6354 | 2.1180 | $-0.0133$ | 0.6493 | 0.6597 |
| DVS-CIFAR10 | Acc $\uparrow$ | 2.0379 | $-0.0693$ | 0.4170 | 2.2000 | $-0.0222$ | 0.4100 | 0.4280 |

In other words, $g_{\mathrm{unit}}$ is the one-point tangent root estimate obtained from the normalized operator alone. This estimator is attractive because it needs only a single Lyapunov evaluation, but it inherits any slope mismatch caused by gain-dependent nonlinear saturation, leak/adaptation, or task-induced state redistribution.

The two-point estimator instead probes a second nearby gain $g_2$ with exponent $\lambda_2 \triangleq \lambda_{\max}(g_2)$ and fits the local secant slope

$$\widehat{b} = \frac{\lambda_2 - \lambda_1}{\log g_2}, \qquad g_{\mathrm{two}} = \exp\!\left(-\frac{\lambda_1}{\widehat{b}}\right) = g_2^{-\lambda_1/(\lambda_2 - \lambda_1)}. \tag{63}$$

By the mean-value theorem, $\widehat{b} = \phi'(\xi)$ for some $\xi \in (0, \log g_2)$ whenever $\phi$ is differentiable on the probed interval, so the two-point estimator is a bona fide local root estimate rather than an ad hoc post-training fit. It is exact whenever $\phi$ is affine in $x = \log g$ on that interval, and more generally its error is first-order controlled by the local curvature term in Eq. (60). This is the reason for the extra probe: it corrects the most important failure mode of the unit estimator, namely task-dependent deviation of the true slope from 1, at the cost of only one additional pre-training QR pass.

Table 6 shows two distinct facts. First, the selected pre-training exponents remain slightly negative for both direct estimators. This is intentional: the initializer aims for the stable side of the boundary, not exact zero, because finite-time estimation noise and trainable-model mismatch make a small safety margin preferable. Second, the two-point estimator usually, but not uniformly, moves $\lambda_{\mathrm{pre}}$ closer to zero than the unit estimator. What matters empirically is that the end-task metric gap remains modest on most tasks even when the pre-training exponent is only approximately placed: the clearest outlier is still Row-CIFAR10, whereas the other benchmarks usually stay within a small neighborhood of the scan baseline. This is why we describe the direct method as a practical replacement for repeated sweeps rather than a universal improvement over every scanned optimum.

The apparent "$g \approx 1$ in the sweep but $g \approx 2$ in the direct table" discrepancy is only a coordinate change. The sweep plots use a raw multiplier $\tilde{g}$ on the stored matrix $\widetilde{\mathbf{W}}_{hh}$, whereas the direct table uses the absolute normalized gain in $\mathbf{W}_{hh} = g\overline{\mathbf{W}}_{hh}$ with $\|\overline{\mathbf{W}}_{hh}\|_{\mathrm{op}} = 1$. The relation is $g = \tilde{g}\,\|\widetilde{\mathbf{W}}_{hh}\|_{\mathrm{op}}$, so a sweep crossing near $\tilde{g} \in [0.6, 1.0]$ can map to an absolute normalized gain near 2 after restoring the base operator norm. Across tasks, the remaining variation comes from the task-driven contraction hidden in $\mathbf{D}_t(g)$: nonlinear slopes, leak/adaptation, and state occupancy all change how much gain is needed to approach the same stable boundary. Figures 2 and 3 visualize this behavior for representative RNN and ConvRNN tasks, respectively.

**Why $\lambda_{\max}$ often becomes more negative after training.** Training commonly moves the model away from the exact boundary and toward a task-dependent stable regime. This effect is strongest on Lorenz rollout prediction, where contractive post-training dynamics improve closed-loop stability; it is milder on memory-intensive tasks that still benefit from longer recurrent echoes.

### H.3. Spatial mismatch robustness

To test how strongly COLA depends on the long-range spatial correlation assumption, we perturb the source term by mixing the true source $s_t$ with a random normalized pattern $q_t$ as $(1 - m)s_t + mq_t$, where $m$ controls the mismatch strength.

Table 7 makes the pattern explicit: mild mismatch is often absorbed and can even help slightly on some tasks, whereas

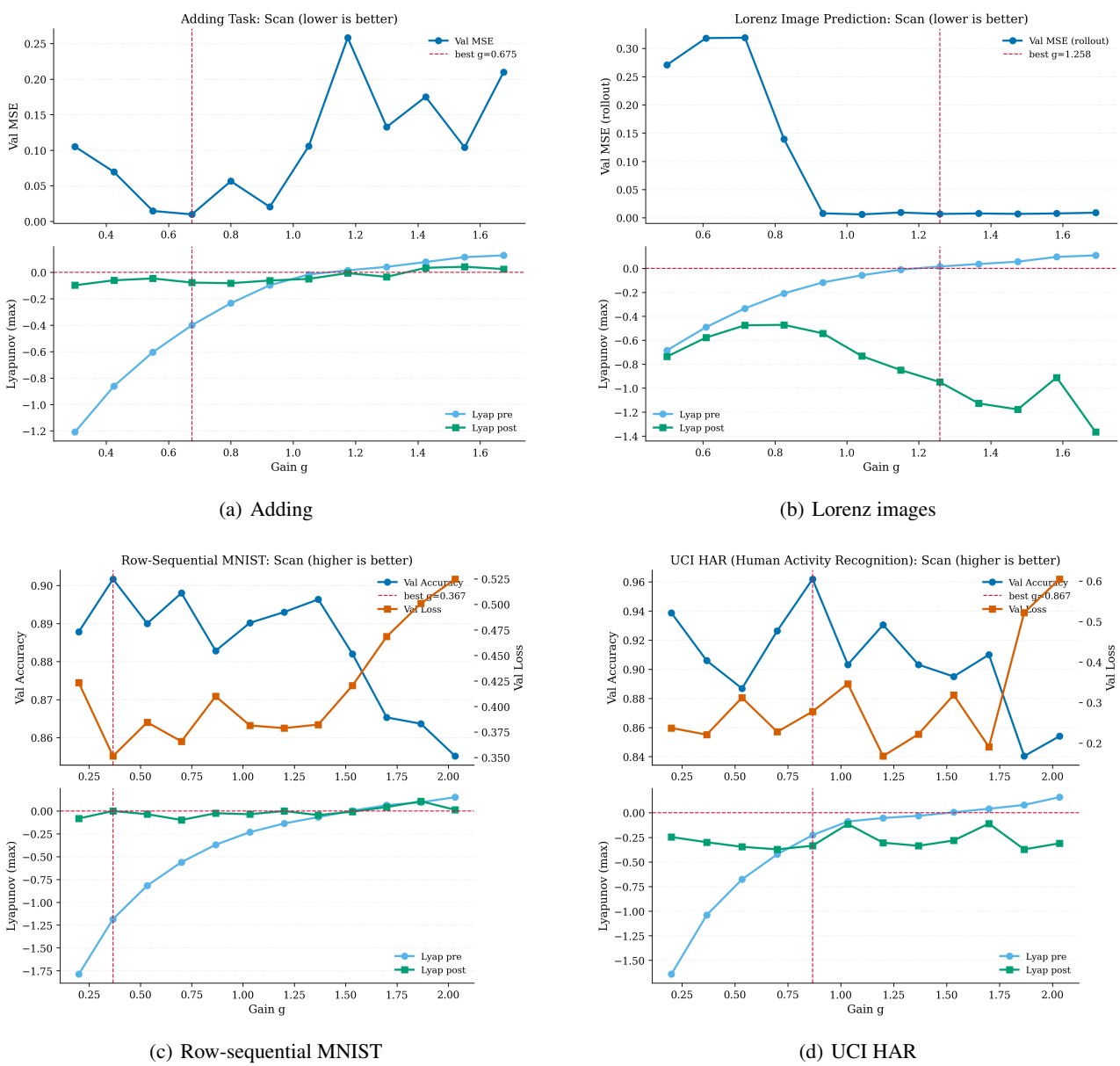

(a) Adding

(b) Lorenz images

(c) Row-sequential MNIST

(d) UCI HAR

*Figure 2.* Representative RNN gain sweeps. Each plot shows the validation metric together with the pre-training and post-training maximum Lyapunov exponent.

strong mismatch degrades performance reliably on the more sensitive benchmarks. This supports the interpretation of long-range spatial correlation as a useful but non-binary approximation.

# I. Task details and hyperparameters

**Purpose.** This appendix collects reproducibility details for Sec. 4: task descriptions, preprocessing, and default hyperparameters.

**Summary tables.** Tables 8 and 9 summarize the default hyperparameters used in the main paper.

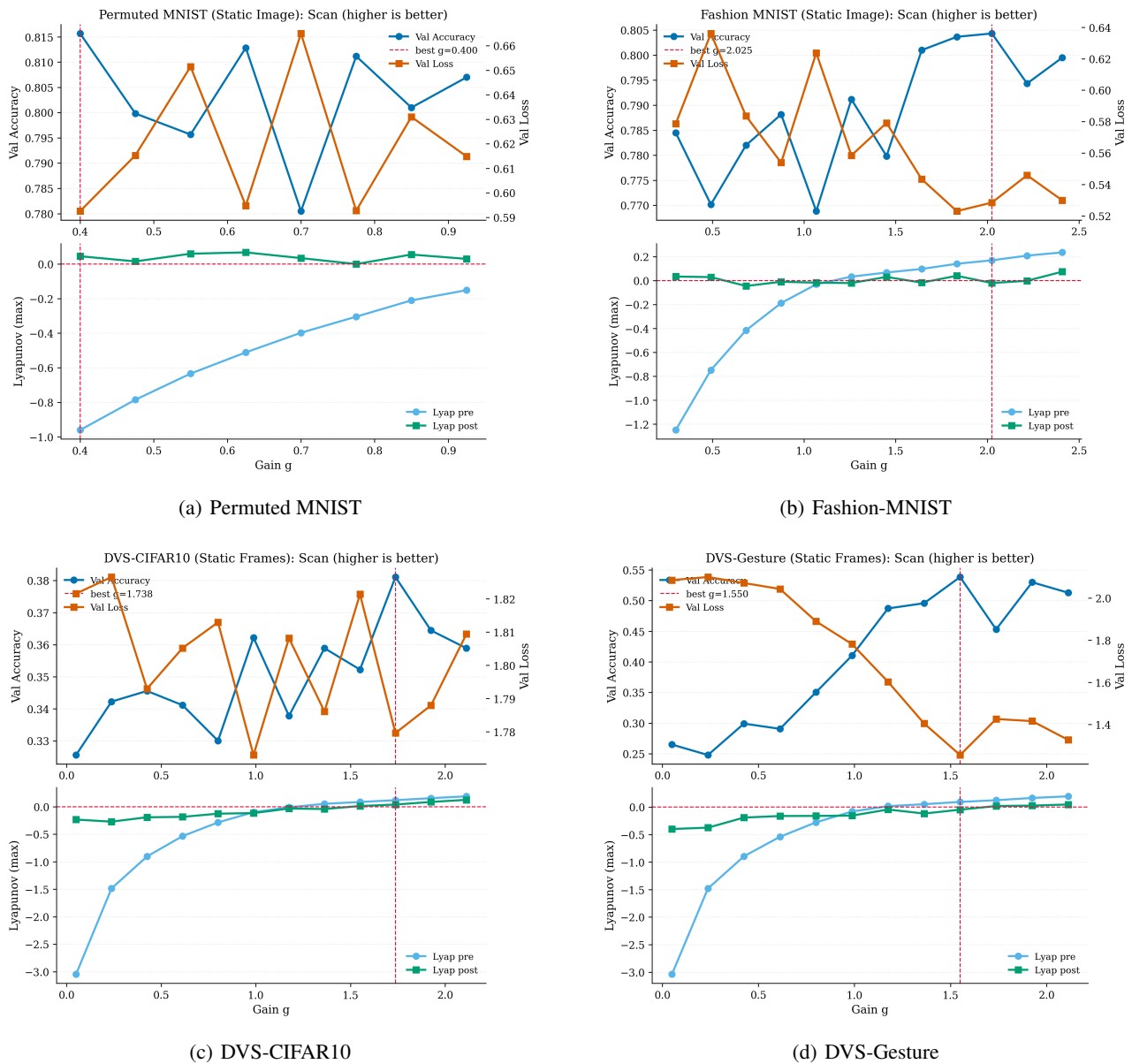

(a) Permuted MNIST

(b) Fashion-MNIST

(c) DVS-CIFAR10

(d) DVS-Gesture

*Figure 3.* Representative ConvRNN gain sweeps. The selected initialization lies near the stable boundary, while post-training dynamics often become more contractive.

*Table 8.* Hyperparameters for RNN tasks. Splits are given as train/val/test sizes (examples for synthetic tasks, images for vision tasks, and tokens for language modeling).

| Task | Type | Metric | Unroll | In/Out | $H$ | Batch | Epochs | Split |
|------|------|--------|--------|--------|-----|-------|--------|-------|
| Adding | Reg. | MSE | 60 | 2/1 | 128 | 128 | 30 | 8000/800/2000 |
| Lorenz images | Reg. | Rollout MSE | 30 | 256/256 | 192 | 64 | 30 | 10000/1000/1000 |
| Row-MNIST | Cls. | Acc | 28 | 28/10 | 128 | 64 | 50 | 60000/6000/10000 |
| Row-CIFAR10 | Cls. | Acc | 32 | 96/10 | 128 | 64 | 100 | 50000/5000/10000 |
| UCI HAR | Cls. | Acc | 128 | 9/6 | 128 | 64 | 50 | 7352/726/2947 |
| PTB-Char LM | LM | Loss | Block 80 | 50/50 | 128 | 64 | 50 | 5101618/399782/449945 |
| WT2-Char LM | LM | Loss | Block 80 | 250/250 | 128 | 64 | 50 | 10886935/1148599/1255101 |

*Table 7.* Spatial mismatch robustness. The table lists the exact endpoint metrics under no mismatch ($m = 0$), mild mismatch ($m = 0.25$), and strong mismatch ($m = 0.75$). For regression, lower is better; for classification, higher is better.

| Task | $m = 0$ | $m = 0.25$ | $m = 0.75$ |
|---|---|---|---|
| Adding (MSE) | 0.0086 | 0.0053 | 0.1088 |
| Row-MNIST (Acc) | 0.9336 | 0.9668 | 0.9023 |
| Row-CIFAR10 (Acc) | 0.4258 | 0.4453 | 0.3320 |
| Lorenz Image (MSE) | 0.0029 | 0.0043 | 0.0086 |
| UCI HAR (Acc) | 0.8359 | 0.8398 | 0.6758 |

*Table 9.* Hyperparameters for ConvRNN refinement tasks. Splits are given as train/val/test examples.

| Task | Steps | Encoder ch. | Hidden ch. | Kernel | Batch | Epochs | Split |
|---|---|---|---|---|---|---|---|
| Permuted MNIST | 12 | 16, 32 | 128 | 3 | 64 | 50 | 60000/6000/10000 |
| Fashion-MNIST | 12 | 16, 32 | 128 | 3 | 64 | 50 | 60000/6000/10000 |
| DVS-CIFAR10 | 12 | 32, 64 | 256 | 3 | 64 | 50 | 9000/900/1000 |
| DVS-Gesture | 12 | 32, 64 | 256 | 3 | 64 | 100 | 1077/107/264 |

**Task descriptions.** **Adding task.** The classic adding problem (Arjovsky et al., 2016) is used to probe precise long-range memory retention without gating. Each input is a length-60 sequence with two channels $(v_t, m_t)$, where the target is the sum of two masked values $v_t$ that are widely separated in time. This task is challenging for vanilla RNNs trained with BPTT because gradients must propagate across a long lag.

**Lorenz image prediction (rollout).** Trajectories of the Lorenz system (Lorenz, 2017) are rendered into $16 \times 16$ frames, and training uses teacher forcing. The key evaluation is closed-loop rollout: the model must generate a coherent trajectory without ground-truth feedback, a setting that is highly sensitive to dynamical instability.

**Row-sequential MNIST and CIFAR-10.** Static images are converted into sequences by feeding one image row per time step and predicting the class from the final-step logits. For MNIST (Lecun et al., 1998), each step is a length-28 row, giving 28 steps total; for CIFAR-10 (Krizhevsky, 2009), each step is a length-96 row (32 pixels with 3 channels), giving 32 steps total.

**UCI HAR.** UCI HAR (Anguita et al., 2013) is a human activity recognition dataset with 128-step sensor sequences and 9 input channels, labeled into 6 activities.

**PTB-Char and WikiText-2 Char language modeling.** Character-level language models are trained by predicting the next character given the previous context. PTB-Char is derived from the Penn Treebank (Marcus et al., 1993); WikiText-2 Char uses the WikiText-2 corpus (Merity et al., 2017). Training uses fixed-length blocks (default 80) and test cross-entropy loss is evaluated.

**ConvRNN static-image refinement.** For Permuted MNIST and Fashion-MNIST (Xiao et al., 2017), classification is treated as recurrent refinement: the same image is provided at every step and the ConvRNN is unrolled for a fixed number of steps (default 12). Accuracy is taken from the final-step logits.

**Event-based tasks.** For CIFAR10-DVS (Li et al., 2017) and DVS Gesture (Amir et al., 2017), event streams are discretized into static frame stacks using a fixed number of time bins and the same recurrent refinement pipeline is applied. Ten time bins are used for CIFAR10-DVS and 20 time bins for DVS Gesture; event polarity is kept, yielding $2B$ input channels for $B$ time bins, and spatial resolution is downsampled by a factor of 4.

## J. ConvRNN extension and benchmarks

### J.1. Convolutional extension: ConvRNN

The same construction extends to convolutional recurrent networks by treating each hidden channel as a unit and maintaining per-channel scalar statistics. A convolutional neural network (CNN) encoder maps an input image $\mathbf{I}$ to a feature map

$$\mathbf{f} = \text{Enc}(\mathbf{I}). \tag{64}$$

*Table 10.* ConvRNN refinement benchmarks (mean ± std over seeds 42/43/44). Best among the methods shown is bold.

| Task | BPTT | TBPTT-1 | TBPTT-10 | e-prop | FPTT | UORO | **COLA** |
|---|---|---|---|---|---|---|---|
| Fashion-MNIST | **0.8730** | 0.8347 | 0.8685 | 0.8506 | 0.7941 | 0.5616 | 0.8467 |
| Acc↑ | ± **0.0031** | ± 0.0066 | ± 0.0020 | ± 0.0084 | ± 0.0070 | ± 0.0124 | ± 0.0014 |
| Permuted MNIST | 0.9325 | 0.9114 | **0.9340** | 0.9174 | 0.8764 | 0.3918 | 0.9122 |
| Acc↑ | ± 0.0030 | ± 0.0046 | ± **0.0033** | ± 0.0067 | ± 0.0071 | ± 0.0553 | ± 0.0027 |
| DVS-Gesture | 0.6678 | 0.6215 | **0.6852** | 0.6551 | 0.6377 | 0.1493 | 0.6597 |
| Acc↑ | ± 0.0140 | ± 0.0309 | ± **0.0145** | ± 0.0244 | ± 0.0231 | ± 0.0874 | ± 0.0035 |
| DVS-CIFAR10 | 0.4540 | 0.4300 | **0.4680** | 0.4263 | 0.3697 | 0.1360 | 0.4410 |
| Acc↑ | ± 0.0056 | ± 0.0213 | ± **0.0078** | ± 0.0105 | ± 0.0093 | ± 0.0144 | ± 0.0141 |

ConvRNN maintains a hidden feature map $\mathbf{H}_t \in \mathbb{R}^{C_h \times H' \times W'}$; global average pooling (GAP) over spatial positions is used to obtain a channel vector readout:

$$\mathbf{X}_t = \mathrm{Conv}_{hh}(\mathbf{H}_{t-1}) + \mathrm{Conv}_{xh}(\mathbf{f}) + \mathbf{b}_h, \tag{65}$$
$$\mathbf{H}_t = \tanh(\mathbf{X}_t),$$
$$\mathbf{z}_t = \mathrm{GAP}(\mathbf{H}_t) \in \mathbb{R}^{C_h},$$
$$\mathbf{o}_t = \mathbf{W}_{hy}\mathbf{z}_t + \mathbf{b}_y. \tag{66}$$

Losses and step weights follow Eq. (3).

Let $\mathbf{U}_t = 1 - \mathbf{H}_t \odot \mathbf{H}_t$ denote elementwise slopes. Let $\mathbf{B}_t$ be the spatial broadcast of $\mathbf{W}_{hy}^{\top} \frac{\partial \mathcal{L}_t}{\partial \mathbf{o}_t} \in \mathbb{R}^{C_h}$. We maintain a per-channel loop gain $\boldsymbol{\mu} \in \mathbb{R}^{C_h}$ (broadcast spatially). The teaching signal is

$$\widetilde{\boldsymbol{\Delta}}_t = (\mathbf{U}_t \odot \mathbf{B}_t) \oslash (\mathbf{1} - \boldsymbol{\mu} \odot \mathbf{U}_t), \tag{67}$$

with the same denominator safeguards as in Eq. (31). Kernel updates use standard cross-correlation between $\widetilde{\boldsymbol{\Delta}}_t$ and the corresponding input maps, and readout updates use the exact instantaneous gradient with respect to logits. Per-channel scalar statistics are updated using the same EMA-style estimators as in Sec. 3.3.4; in particular, $\alpha$ is estimated from $\widetilde{\boldsymbol{\Delta}}_t$ via the AR(1) EMA with sufficient statistics aggregated across batch and spatial positions.

If the encoder is trainable, gradients are backpropagated through the encoder within each step, using the instantaneous gradient on $\mathbf{f}$ induced by $\mathrm{Conv}_{xh}$.

## J.2. ConvRNN tasks.

We extend the evaluation to convolutional recurrent architectures for iterative vision tasks. The models employ a two-layer strided CNN encoder feeding a 2D ConvRNN core with $\tanh$ nonlinearity and $128$ hidden channels, where both input-to-hidden and hidden-to-hidden transformations are convolutional. Training uses cross-entropy with labels repeated across $T=12$ steps, and performance is measured by final-step accuracy. As shown in Table 10, COLA remains a competitive online alternative, but global BPTT or long-window TBPTT still achieve the strongest final accuracies.

**Static-image refinement.** On Fashion-MNIST and Permuted MNIST, the network performs recurrent refinement (Liao & Poggio, 2016) on persistent sensory input. COLA stays close to e-prop and clearly above UORO, suggesting that the per-channel loop-gain statistics of Section J.1 capture useful spatial dependencies. The remaining gap to TBPTT-10 indicates that these tasks still benefit from more exact multi-step spatial credit assignment.

**Event-based classification.** On DVS-Gesture and DVS-CIFAR10, COLA reaches $65.97\%$ and $44.10\%$, respectively. It remains above TBPTT-1 and competitive with e-prop, but again trails BPTT and TBPTT-10. This is consistent with the main claim of the paper: COLA scales to high-dimensional recurrent feature maps, but the empirical benefit is strongest when long-horizon recurrent stability matters more than exact offline unrolling.

# K. SNN extension and benchmarks

## K.1. Why these spiking cells remain compatible with COLA

The relevant structural requirement is not literally "one scalar state variable," but one recurrently coupled transport chain through which long-range credit must pass. RLIF has only the membrane-potential chain. RadLIF adds a neuron-local adaptation trace, and EGRU adds gates and thresholded event emission around a continuous hidden state. In our implementation, however, the recurrently coupled pathway to which COLA is applied remains the membrane/hidden chain: $\mathbf{v}_t$ for RLIF and RadLIF, and $\mathbf{h}_t$ for EGRU. Extra traces and gates therefore modify the local Jacobian seen by this chain rather than creating an LSTM-style second global memory route. This scope is complementary to online SNN methods such as OSTL and the recent model-agnostic linear-memory rule of Wang et al. (Bohnstingl et al., 2022; Wang et al., 2026): those works show that online credit assignment can remain effective in spiking systems, while our focus here is the narrower question of when the COLA scalar closure continues to track the recurrently coupled state chain.

## K.2. Cell equations

The benchmark suite uses N-MNIST (Orchard et al., 2015), SHD (Cramer et al., 2022), and DVS-Gesture (Amir et al., 2017). For all cells, the recurrent matrix is scaled as $\mathbf{W}_{\mathrm{rec}}(g) = g\,\widetilde{\mathbf{W}}_{\mathrm{rec}}$ and applied together with a fixed binary mask $\mathbf{M}$.

**RLIF.** For input $\mathbf{x}_t$, membrane state $\mathbf{v}_t$, and binary spike output $\mathbf{s}_t$, RLIF evolves as

$$\mathbf{i}_t = \mathbf{W}_{\mathrm{in}}\mathbf{x}_t + \big(\mathbf{W}_{\mathrm{rec}}(g) \odot \mathbf{M}\big)\mathbf{s}_{t-1}, \tag{68}$$

$$\mathbf{v}_t = \boldsymbol{\alpha} \odot \mathbf{v}_{t-1} - \boldsymbol{\alpha} \odot \mathbf{s}_{t-1} + \big(\mathbf{1} - \boldsymbol{\alpha}\big) \odot \mathbf{i}_t, \tag{69}$$

$$\mathbf{s}_t = H\big(\mathbf{v}_t - \boldsymbol{\vartheta}\big). \tag{70}$$

COLA is applied to the recurrently coupled membrane pathway $\mathbf{v}_t$.

**RadLIF.** RadLIF augments RLIF with a neuron-local adaptation current $\mathbf{w}_t$:

$$\mathbf{w}_t = \boldsymbol{\beta} \odot \mathbf{w}_{t-1} + \mathbf{a} \odot \mathbf{v}_{t-1} + \mathbf{b} \odot \mathbf{s}_{t-1}, \tag{71}$$

$$\mathbf{i}_t = \mathbf{W}_{\mathrm{in}}\mathbf{x}_t + \big(\mathbf{W}_{\mathrm{rec}}(g) \odot \mathbf{M}\big)\mathbf{s}_{t-1} - \mathbf{w}_t, \tag{72}$$

$$\mathbf{v}_t = \boldsymbol{\alpha} \odot \mathbf{v}_{t-1} - \boldsymbol{\alpha} \odot \mathbf{s}_{t-1} + \big(\mathbf{1} - \boldsymbol{\alpha}\big) \odot \mathbf{i}_t, \tag{73}$$

$$\mathbf{s}_t = H\big(\mathbf{v}_t - \boldsymbol{\vartheta}\big). \tag{74}$$

The adaptation trace $\mathbf{w}_t$ changes the local slope and effective contraction, but recurrent coupling across neurons still enters through the membrane chain $\mathbf{v}_t$.

**EGRU.** On DVS-Gesture we also evaluate an event-gated GRU branch. For continuous hidden state $\mathbf{h}_t$, update gate $\mathbf{z}_t$, reset gate $\mathbf{r}_t$, and pre-threshold candidate state $\mathbf{c}_t$,

$$\mathbf{z}_t = \sigma\!\left(\mathbf{W}_z \begin{bmatrix} \mathbf{x}_t \\ \mathbf{h}_{t-1} \end{bmatrix} + \mathbf{b}_z\right), \quad \mathbf{r}_t = \sigma\!\left(\mathbf{W}_r \begin{bmatrix} \mathbf{x}_t \\ \mathbf{h}_{t-1} \end{bmatrix} + \mathbf{b}_r\right), \tag{75}$$

$$\widetilde{\mathbf{g}}_t = \mathbf{W}_{hx}\mathbf{x}_t + \big(\mathbf{W}_{hh}\mathbf{h}_{t-1}\big) \odot \mathbf{r}_t + \mathbf{b}_h, \quad \mathbf{g}_t = \tanh(\widetilde{\mathbf{g}}_t), \tag{76}$$

$$\mathbf{c}_t = \mathbf{z}_t \odot \mathbf{h}_{t-1} + \big(\mathbf{1} - \mathbf{z}_t\big) \odot \mathbf{g}_t, \tag{77}$$

$$\mathbf{e}_t = H\big(\mathbf{c}_t - \boldsymbol{\vartheta}\big), \quad \mathbf{m}_t = \mathbf{e}_t \odot \mathbf{c}_t, \quad \mathbf{h}_t = \mathbf{c}_t - \mathbf{e}_t \odot \boldsymbol{\vartheta}. \tag{78}$$

The recurrently coupled state remains $\mathbf{h}_t$; the event gate $\mathbf{e}_t$ and emitted magnitude $\mathbf{m}_t$ determine what is propagated forward and read out.

**Readouts.** For RLIF and RadLIF, the last-layer spikes are passed through a low-pass readout,

$$\mathbf{y}_t = \mathbf{W}_{\mathrm{out}}\mathbf{s}_t^{(L)},$$

$$\mathbf{u}_t = \boldsymbol{\alpha}_{\mathrm{out}} \odot \mathbf{u}_{t-1} + \big(\mathbf{1} - \boldsymbol{\alpha}_{\mathrm{out}}\big) \odot \mathbf{y}_t,$$

$$\mathbf{o} = \sum_{t=1}^{T} \mathbf{u}_t. \tag{79}$$

For EGRU, the emitted event state is first smoothed by a trace

$$\boldsymbol{\tau}_t = \alpha_{\mathrm{tr}}\boldsymbol{\tau}_{t-1} + \big(1 - \alpha_{\mathrm{tr}}\big)\mathbf{m}_t^{(L)}, \qquad \mathbf{o}_t^{\mathrm{egru}} = \mathbf{W}_{\mathrm{out}}^{\mathrm{egru}}\boldsymbol{\tau}_t + \mathbf{b}_{\mathrm{out}}. \tag{80}$$

### K.3. Benchmark interpretation

Table 3 in the main text includes only the directly comparable three-seed pairings used in the paper narrative. That paired view is the strongest claim we can make, because it holds architecture and training budget fixed. The outcome is mixed rather than uniformly positive: COLA wins on N-MNIST/RLIF and on both SHD settings, remains close on DVS-Gesture/EGRU, and is not uniformly favorable across all cell-dataset pairs. This is the right interpretation boundary for the SNN extension.

Table 11 gives the complete benchmark alignment used for the broader SNN comparison. These extra rows are useful for scope, but they should not be over-interpreted as controlled head-to-head evidence because many literature baselines differ in cell type, readout, preprocessing, or online/offline supervision protocol. Even with that caveat, the qualitative picture is stable: COLA remains viable on RLIF/RadLIF/EGRU-style recurrence, yet the gain is clearly architecture dependent rather than universal.

## L. Supplementary experiments

**Scope.** This appendix presents learning curves, pairwise comparisons against approximate-RTRL baselines, one-shot initialization cost, tuned-BPTT fairness checks, fast-target stress tests, and architectural pilots on more strongly coupled recurrence. Throughout, the intended interpretation is conservative: COLA is strongest on long-horizon settings where recurrence remains concentrated in one coupled state chain, competitive but not uniformly dominant elsewhere, and near-criticality should be read as an initialization scaffold rather than a persistent training law.

### L.1. Learning curves

Figure 4 shows mean validation-accuracy trajectories with $\pm 1$ standard-deviation bands over seeds 42/43/44. These plots are regenerated directly from the stored JSON histories used for the final multiseed tables.

### L.2. Profiled step cost

This subsection presents explicit profiler measurements of engineering cost on the same VGPU-32GB machine and profiling path used for the archived reruns. We use per-step wall-clock time, because chunked or truncated methods do not perform one optimizer update at every recurrent step, and this normalization provides a fair comparison across BPTT, truncated BPTT, forward-mode methods, and approximate-RTRL rules. The table also lists per-step FLOPs and peak update CUDA memory in MiB.

*Table 12.* Per-step profiled update cost on the seven RNN benchmarks under a shared profiling protocol on VGPU-32GB.

| Task | Time (COLA / BPTT) | FLOPs (COLA / BPTT) | Peak CUDA delta MiB (COLA / BPTT) |
|---|---|---|---|
| Adding | 0.000123 / 0.000065 | 8.80M / 12.78M | 0.97 / 12.25 |
| Lorenz Image | 0.000132 / 0.000075 | 41.27M / 45.53M | 5.19 / 12.38 |
| PTB Char | 0.000154 / 0.000077 | 8.42M / 10.39M | 2.51 / 11.92 |
| Row-CIFAR10 | 0.000115 / 0.000062 | 7.96M / 9.88M | 1.36 / 9.49 |
| Row-MNIST | 0.000114 / 0.000061 | 5.72M / 7.64M | 0.76 / 9.32 |
| UCI HAR | 0.000120 / 0.000060 | 4.90M / 6.88M | 0.95 / 12.60 |
| WikiText-2 Char | 0.000228 / 0.000176 | 87.81M / 89.46M | 43.07 / 32.61 |

Table 12 summarizes the seven RNN measurements. BPTT is still faster in raw wall-clock time on these small RNNs, but COLA usually reduces per-step FLOPs and drastically lowers peak update memory. The only clear exception is WikiText-2, where the character-level vocabulary expansion makes COLA's lightweight-state advantage less visible in peak CUDA memory. The main claim in the paper is therefore not a blanket speedup claim; it is a memory-and-online-efficiency claim under matched training objectives.

For Figure 1, Table 13 summarizes the common row-sequential classification regime over Row-MNIST, Row-CIFAR10, and UCI HAR for every method that appears in the main RNN table. The accuracy coordinate is the arithmetic mean of the

*Table 11.* Full SNN classification comparison aligned with the source benchmark papers. COLA rows show three-seed mean ± sample standard deviation.

| Dataset | Method | Cell | Test acc. |
|---|---|---|---|
| N-MNIST | Offline / BPTT | RLIF | 98.33 ± 0.04% |
| | Online / pp-prop | RLIF | 98.25 ± 0.03% |
| | Ours / COLA | RLIF | **98.54 ± 0.03%** |
| | Offline / BPTT | RadLIF | 98.29 ± 0.02% |
| | Online / pp-prop | RadLIF | 98.40 ± 0.03% |
| | Ours / COLA | RadLIF | 93.56 ± 0.12% |
| | Online / ETLP | RLIF | 94.30% |
| | Online / e-prop | RLIF | 97.90% |
| | Online / OSTL | sSNU | 96.80 ± 0.17% |
| SHD | Offline / BPTT | RLIF | 94.01 ± 0.12% |
| | Online / pp-prop | RLIF | 93.93 ± 0.28% |
| | Ours / COLA | RLIF | **94.07 ± 0.45%** |
| | Offline / BPTT | RadLIF | 94.72 ± 1.06% |
| | Online / pp-prop | RadLIF | 95.33 ± 0.11% |
| | Ours / COLA | RadLIF | **95.37 ± 0.13%** |
| | Online / ETLP | RLIF | 78.71 ± 1.49% |
| | Online / e-prop | RLIF | 80.79 ± 0.39% |
| | Online / OTPE | LIF | 76.70 ± 0.70% |
| | Online / e-prop | TC-RLIF | 80.57% |
| | Online / OSTTP | sNU | 77.33 ± 0.8% |
| | Online / S-TLLR | RLIF | 78.24 ± 1.84% |
| | Offline / EventProp | RLIF | 93.50 ± 0.70% |
| | Offline / BPTT | DCLS-Delays | 95.07 ± 0.24% |
| | Offline / BPTT | RadLIF | 94.62% |
| DVS-Gesture | Offline / BPTT | RLIF | 93.97 ± 0.15% |
| | Online / pp-prop | RLIF | 94.26 ± 0.32% |
| | Ours / COLA | RadLIF | 93.38 ± 0.33% |
| | Offline / BPTT | EGRU | 97.45 ± 0.27% |
| | Online / pp-prop | EGRU | 97.29 ± 0.16% |
| | Ours / COLA | EGRU | 97.19 ± 0.32% |
| | Online / FPTT | RLIF | 92.13 ± 0.87% |
| | Online / FPTT | CNN | 97.22% |
| | Online / OTTT | VGG-11 | 96.88% |
| | Offline / BPTT | STS-ResNet | 96.7% |
| | Offline / BPTT | STL-SNN | 97.01 ± 0.23% |
| | Offline / BPTT | PLIF / VGGSNN | 97.57% |

corresponding entries in Table 2, while the cost coordinates are averaged from the matched profiler outputs over the same three tasks.

*Table 13.* Three-task averages used in the empirical bottom panels of Figure 1.

| Method | Avg. acc. | Time / step | MFLOPs / step | Mem. MiB |
|---|---|---|---|---|
| BPTT | 0.7712 | 0.000061 | 8.14 | 10.47 |
| TBPTT-1 | 0.7218 | 0.000101 | 6.09 | 0.31 |
| TBPTT-10 | 0.7764 | 0.000065 | 7.95 | 0.71 |
| e-prop | 0.7238 | 0.000136 | 6.15 | 0.41 |
| FPTT | 0.7290 | 0.001021 | 7.35 | 0.57 |
| UORO | 0.3613 | 0.000221 | 13.99 | 1.01 |
| SnAp-1 | 0.7216 | 0.000126 | 6.19 | 14.02 |
| COLA | 0.7551 | 0.000117 | 6.19 | 1.02 |

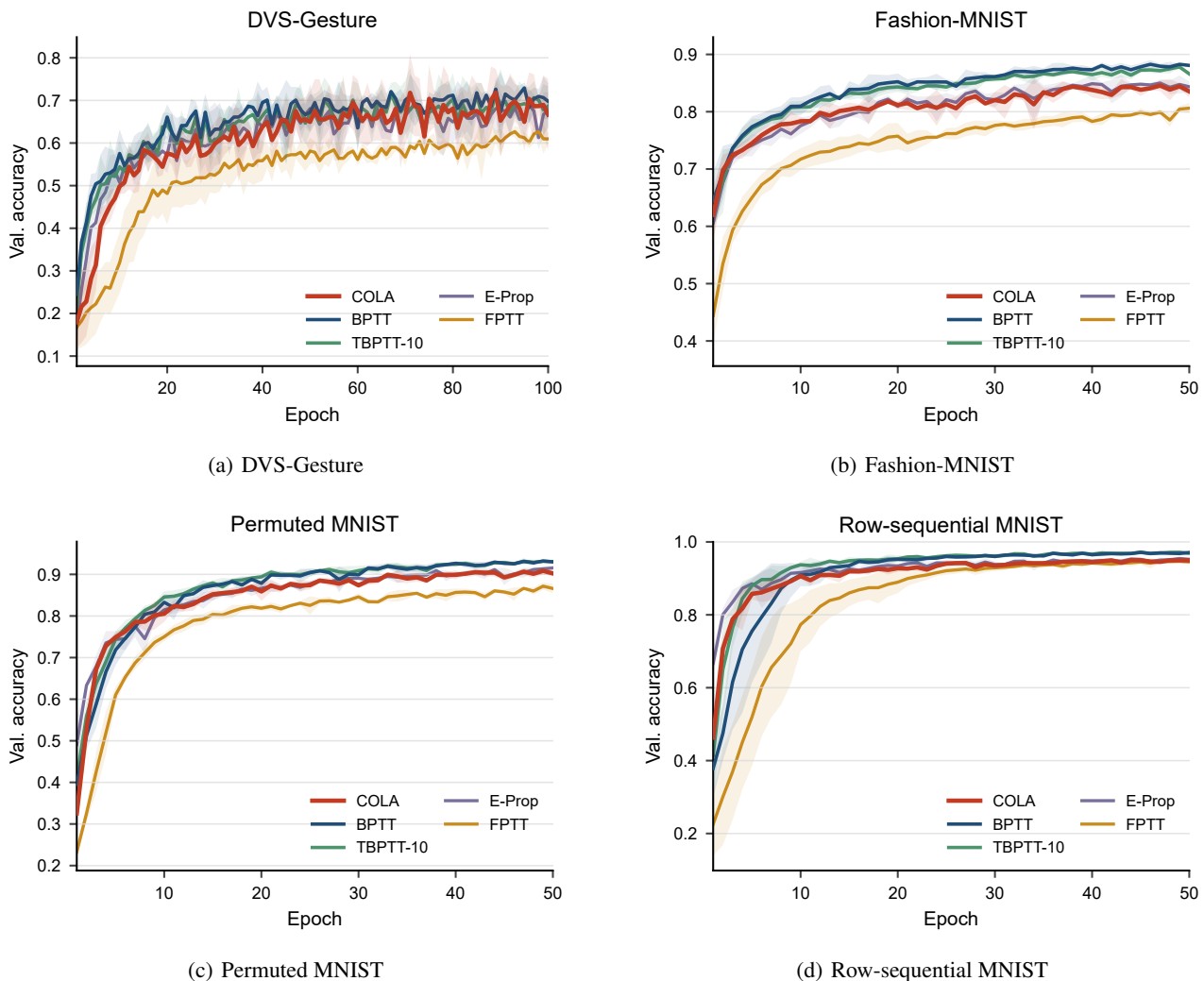

*Figure 4.* Learning curves (validation accuracy) across four benchmarks. Each line shows the seed mean and the shaded band indicates $\pm 1$ standard deviation.

## L.3. EMA hyperparameter sensitivity

We performed dense local sweeps of the COLA EMA hyperparameters around the default setting $\rho_\alpha = 0.995$ and $W_\lambda = 50$, using

$$\rho_\alpha \in \{0.95, 0.97, 0.98, 0.99, 0.995, 0.999\}, \qquad W_\lambda \in \{10, 20, 30, 50, 75, 100, 150\}.$$

In these runs, the recurrent gain is held fixed at the scan-selected baseline for each task, so the sweep isolates the EMA continuation statistics rather than conflating them with a new gain search. Table 14 prints the complete dense archive across all 11 formal tasks; the default configuration appears in the rows $\rho_\alpha = 0.995$ and $W_\lambda = 50$. These sensitivity rows are single-archive reruns for local hyperparameter comparisons, not the multiseed averages reported in the main RNN and ConvRNN benchmark tables.

Table 14 shows that the sensitivity remains task dependent but usually modest around the default. UCI HAR is the clearest sensitive case in the archive: the best $\rho_\alpha$ setting (0.98) improves accuracy from 0.8649 to 0.8839 (+0.0190), while the best window setting (100) reaches 0.8734 (+0.0085). DVS-Gesture is a moderate-sensitivity case: the best $\rho_\alpha$ setting (0.97) improves accuracy from 0.6493 to 0.6701 (+0.0208), and the best window setting (10) reaches 0.6667 (+0.0174).

At the other extreme, PTB-Char is flat to the shown precision across the entire parameter grid. Under the fixed-gain setting,

this phenomenon can be explained as follows: at the selected PTB gain, the projected loop-gain term reaches the same stable saturation cap, thereby leaving the online weight update trajectory unchanged across the entire grid. WikiText-2 Char shows only sub-$10^{-3}$ variation (mainly in the $W_\lambda$ sweep), and the Lorenz task is also nearly flat: the full parameter sweep changes the rollout MSE only from $0.004837$ to $0.005222$. The remaining tasks lie between these two extremes, with most improvements from retuning staying small relative to the default setting.

*Table 14.* Complete dense EMA sensitivity sweeps across all 11 archived tasks. For the $\rho_\alpha$ sweep, $W_\lambda{=}50$ is fixed; for the $W_\lambda$ sweep, $\rho_\alpha{=}0.995$ is fixed. Lower is better for MSE/loss and higher is better for accuracy.

| Sweep | Value | Adding MSE↓ | Lorenz MSE↓ | PTB-Char Loss↓ | WT2-Char Loss↓ | Row-MNIST Acc↑ | Row-CIFAR10 Acc↑ | UCI HAR Acc↑ | Fashion-MNIST Acc↑ | Permuted MNIST Acc↑ | DVS-Gesture Acc↑ | DVS-CIFAR10 Acc↑ |
|---|---|---|---|---|---|---|---|---|---|---|---|---|
| $\rho_\alpha$ | 0.95 | 0.007592 | 0.005003 | 1.841630 | 2.406139 | 0.9546 | 0.4380 | 0.8717 | 0.8380 | 0.9181 | 0.6632 | 0.4350 |
| $\rho_\alpha$ | 0.97 | 0.006460 | 0.004842 | 1.841630 | 2.406139 | 0.9519 | 0.4423 | 0.8392 | 0.8395 | 0.9154 | 0.6701 | 0.4330 |
| $\rho_\alpha$ | 0.98 | 0.005818 | 0.004837 | 1.841630 | 2.406139 | 0.9539 | 0.4321 | 0.8839 | 0.8339 | 0.9129 | 0.6493 | 0.4500 |
| $\rho_\alpha$ | 0.99 | 0.012664 | 0.004838 | 1.841630 | 2.406139 | 0.9522 | 0.4352 | 0.8616 | 0.8361 | 0.9240 | 0.6458 | 0.4340 |
| $\rho_\alpha$ | **0.995** | 0.009118 | 0.004885 | 1.841630 | 2.406139 | 0.9553 | 0.4415 | 0.8649 | 0.8409 | 0.9219 | 0.6493 | 0.4320 |
| $\rho_\alpha$ | 0.999 | 0.006389 | 0.005045 | 1.841630 | 2.406139 | 0.9537 | 0.4384 | 0.8728 | 0.8430 | 0.9191 | 0.6632 | 0.4400 |
| $W_\lambda$ | 10 | 0.007691 | 0.005222 | 1.841630 | 2.405601 | 0.9517 | 0.4417 | 0.8490 | 0.8364 | 0.9183 | 0.6667 | 0.4160 |
| $W_\lambda$ | 20 | 0.008546 | 0.004887 | 1.841630 | 2.406116 | 0.9521 | 0.4438 | 0.8371 | 0.8370 | 0.9140 | 0.6528 | 0.4370 |
| $W_\lambda$ | 30 | 0.003934 | 0.004894 | 1.841630 | 2.406128 | 0.9470 | 0.4346 | 0.8500 | 0.8357 | 0.9168 | 0.6424 | 0.4400 |
| $W_\lambda$ | **50** | 0.009118 | 0.004885 | 1.841630 | 2.406139 | 0.9553 | 0.4415 | 0.8649 | 0.8409 | 0.9219 | 0.6493 | 0.4320 |
| $W_\lambda$ | 75 | 0.008490 | 0.005033 | 1.841630 | 2.406140 | 0.9550 | 0.4389 | 0.8432 | 0.8435 | 0.9208 | 0.6562 | 0.4280 |
| $W_\lambda$ | 100 | 0.009679 | 0.005041 | 1.841630 | 2.406140 | 0.9558 | 0.4377 | 0.8734 | 0.8417 | 0.9186 | 0.6632 | 0.4150 |
| $W_\lambda$ | 150 | 0.011336 | 0.005032 | 1.841630 | 2.406140 | 0.9553 | 0.4420 | 0.8602 | 0.8384 | 0.9227 | 0.6597 | 0.4290 |

The practical conclusion is unchanged: $(0.995, 50)$ is a robust starting point rather than a knife-edge optimum.

### L.4. Initialization overhead

The direct initializer is intended to replace repeated task-specific sweeps, so its wall-clock overhead must be small relative to training. The relevant claim is therefore not that full COLA training is faster than BPTT, but that the extra near-critical estimation step is a bounded one-shot cost. We measure the marginal cost after the task configuration and Lyapunov driver are fixed, because that is the quantity that directly substitutes for brute-force gain sweeps. Across all 11 formal tasks, the median overhead is $0.163\%$ for the unit estimator and $0.317\%$ for the two-point estimator. Table 15 gives the full RNN+ConvRNN breakdown.

Across the table, most tasks stay around $1\%$ of training time or lower, with DVS-Gesture as the clear heaviest case because its Lyapunov driver is substantially larger than the other refinement inputs. The substantive point is scale separation: one or two short pre-training Lyapunov evaluations are still far cheaper than a full gain scan and remain modest relative to end-to-end training.

*Table 15.* Wall-clock overhead of direct near-critical initialization on the 11 RNN and ConvRNN tasks.

| Task | unit (s) | two-point (s) | train (s) | unit/train | two/train |
|---|---|---|---|---|---|
| Adding | 0.249 | 0.490 | 12.664 | 1.97% | 3.87% |
| Lorenz Image | 0.228 | 0.399 | 16.732 | 1.36% | 2.38% |
| PTB | 0.299 | 0.588 | 185.376 | 0.16% | 0.32% |
| WikiText-2 | 0.327 | 0.629 | 182.340 | 0.18% | 0.35% |
| Row-MNIST | 0.220 | 0.416 | 135.261 | 0.16% | 0.31% |
| Row-CIFAR10 | 0.265 | 0.504 | 259.940 | 0.10% | 0.19% |
| UCI HAR | 0.497 | 0.942 | 80.740 | 0.62% | 1.17% |
| Fashion-MNIST | 0.189 | 0.401 | 960.621 | 0.02% | 0.04% |
| Permuted MNIST | 0.216 | 0.399 | 1008.623 | 0.02% | 0.04% |
| DVS-Gesture | 2.318 | 4.055 | 60.335 | 3.84% | 6.72% |
| DVS-CIFAR10 | 0.220 | 0.392 | 182.510 | 0.12% | 0.22% |

### L.5. Tuned BPTT fairness

To test whether COLA's gains come only from better initialization, we gave BPTT its own gain-tuning budget on the tasks where it originally trailed most strongly. Only the initial recurrent gain is retuned; architecture, optimizer, learning rate,

batch size, and training budget are held fixed. This isolates the most plausible fairness objection without giving BPTT a broader hyperparameter search than COLA. The tuned BPTT baseline improves, especially on Lorenz, but Table 16 shows that it does not close the gap.

*Table 16.* Tuned-BPTT fairness check. Results are mean $\pm$ std over three seeds.

| Task | Tuned BPTT | Original BPTT | COLA |
|---|---|---|---|
| Adding (MSE) | $0.1204 \pm 0.0057$ | $0.1426 \pm 0.0326$ | $\mathbf{0.0059 \pm 0.0026}$ |
| Lorenz Image (MSE) | $0.0412 \pm 0.0115$ | $0.6841 \pm 0.0873$ | $\mathbf{0.0058 \pm 0.0003}$ |
| PTB (Loss) | $2.8809 \pm 0.0126$ | $2.8695 \pm 0.0203$ | $\mathbf{1.8352 \pm 0.0056}$ |

### L.6. Fast-target stress tests

We also ran a simple fast-target experiment designed to test a direct failure mode of the scalar continuation picture: if one scalar continuation statistic were only tracking a single period, then rapidly oscillating targets should break it quickly. Table 17 therefore focuses on period-3/4/5 cosine tasks. These targets change sign and curvature several times within a short horizon, so they directly test whether the continuation statistic remains useful when the target varies rapidly.

*Table 17.* Fast-target stress tests on period-3/4/5 cosine targets (test MSE; lower is better).

| Task | COLA test MSE | BPTT test MSE |
|---|---|---|
| Period-3 cosine mix | 0.0040142 | 0.19317 |
| Period-4 cosine mix | 0.0002781 | 0.37165 |
| Period-5 cosine mix | 0.0025314 | 0.49698 |

These tasks do not suggest that a fixed scalar $\alpha$ literally encodes the output period. Rather, they show that the continuation statistic can still provide a usable short-horizon descent direction even when the target oscillates rapidly.

### L.7. LSTM pilot

LSTM couples multiple persistent state chains, so we treat it here as an empirical comparison for whether the same initialization-and-local-update recipe remains usable in practice. The point is not to provide a complete LSTM theory, but to check whether the recipe remains numerically serviceable once recurrence is no longer concentrated in a single chain; Table 18 reports the pilot comparison.

*Table 18.* Pilot COLA-LSTM versus BPTT-LSTM comparison. For accuracy, higher is better; for loss/MSE, lower is better.

| Task | Metric | BPTT-LSTM | COLA-LSTM |
|---|---|---|---|
| Adding Task | MSE $\downarrow$ | 0.1251 | $\mathbf{0.0816}$ |
| Lorenz Image | MSE $\downarrow$ | 0.0482 | $\mathbf{0.0027}$ |
| Row-MNIST | Acc $\uparrow$ | $\mathbf{0.9712}$ | 0.9509 |
| Row-CIFAR10 | Acc $\uparrow$ | $\mathbf{0.4736}$ | 0.4513 |
| UCI HAR | Acc $\uparrow$ | $\mathbf{0.9074}$ | 0.8867 |
| PTB Char | Loss $\downarrow$ | 3.0299 | $\mathbf{1.9060}$ |
| WikiText-2 | Loss $\downarrow$ | 3.5302 | $\mathbf{2.3821}$ |

### L.8. Long-horizon gesture stress

We additionally ran a long-horizon gesture experiment. As Table 19 shows, BPTT becomes increasingly unstable as the sequence length grows: on ASRNN the accuracy drops from 79.5% to 66.3%, and on LTC-SNN it falls sharply by length 100 and diverges at lengths 500 and 1000, consistent with instability in long rollouts. Under the same horizon extension, COLA remains numerically stable and stays near the mid-80% range on both non-LSTM architectures.

*Table 19.* Long-horizon gesture stress test (accuracy).

**Credit Assignment via Dynamical Criticality**

| Seq. length | LSTM | ASRNN | | LTC-SNN | |
| --- | --- | --- | --- | --- | --- |
| | BPTT | BPTT | COLA | BPTT | COLA |
| 20 | 86.8% | 79.5% | 81.6% | 89.9% | 86.1% |
| 100 | 87.8% | 77.8% | 86.8% | 44.1% | 86.8% |
| 500 | 88.9% | 70.8% | 85.4% | Diverged | 85.8% |
| 1000 | 89.2% | 66.3% | 86.5% | Diverged | 86.5% |

