# OpenReview forum: "Global Credit Assignment via Dynamical Criticality"
_ICML.cc/2026/Conference — ICML 2026 regular_

### Official Review · Reviewer_zsZB · 2026-02-23

**Soundness:** 2
**Presentation:** 3
**Significance:** 3
**Originality:** 3
**Overall Recommendation:** 5
**Confidence:** 3

**Summary:**

The paper derives an approximation of backpropagation through time (BPTT) for recurrent neural networks where the memory is complexity is twice the number of neurons (meaning, each neuron needs to store only 2 variables). The approximation requires the network to be close to criticality (but sub-critical), and two assumptions that loosely imply that the error sent to each neuron is stable or moves slowly, and that the dependencies between neurons also change slowly. The results are validated over various benchmarks, where their method performs well, even better than BPTT in some examples.

**Compliance With Llm Reviewing Policy:**

Affirmed.

**Final Justification:**

The paper is original, and the authors did address my comments.

**Key Questions For Authors:**

The paper cites Whittington & Bogacz as an example of learning over timescales. As far as I remember, Whittington and Bogacz derive an approximation of classical backprop (not through time). I don't fully see the relevance, could you explain?

**Limitations:**

Yes

**Strengths And Weaknesses:**

Strengths:
- the analysis is relatively straight forward
- the approach is new as far as I know.
- The performance for such memory is very good.

Weaknesses:
- When bringing up the assumptions, the paper does not mention how the assumptions relate to the task. Instead, the author discuss the changes in the "credit" given to each neuron as a function of the dynamics of the network, but the $\delta_i$ depends on both. For example, if a linear RNN was trained to generate outputs with alternating values (-1,1,-1,...) the $\delta_i$ that arrives at each neuron would change sign at every iteration. This would clearly break assumption 1. It is not a fundamental invalidation of the approach, but I think it warrants some discussion.
- In their results, COLA sometimes does better than BPTT, which is rather surprising. The authors offer an explanation "While BPTT and other online baselines frequently diverge because of accumulated errors", but this suggest a flaw in the comparison with BPTT. If the performance of BPTT is damaged by the accumulated errors, it suggest that a sub-critical regime (present in COLA, absent in standard BPTT) is the reason of the good performances for COLA. The authors should test if the sub-criticality combined with BPTT does better or worse by adding BPTT (and perhaps some of its variants) with controlled subcriticality to their baselines.
- This brings me to my key concern: the lack of ablations. I think it should be possible to make partial simplifications and check which components are necessary and/or advantageous.
- Details missing in the evaluation: error bars, clock time.

Literature:
- The paper mentions that the brain is critical or near criticality, but refers to theoretical literature, not to any work that actually tested this. It's particularly surprising because there is a fairly convincing literature showing explicitly that the brain is sub-critical but not far off criticlaity, as in the paper (main reference would be Wilting & Priesemann. "Inferring collective dynamical states from widely unobserved systems." Nature communications 9.1 (2018), but Priesemann has a few works about this).
- Neural networks are initialized at criticality for training. This was not explicitly mentioned, which is rather weird given that this is an important modification of COLA.

---

> ### Author Rebuttal · Authors · 2026-03-31
>
> ### Overview
>
> We test the alternating-sign counterexample, give `BPTT` its own tuning budget, redesign gain selection into a direct initializer, add profiling, and tighten the neuroscience framing.
>
> ---
>
> ### 1. "Assumptions ... alternating-sign outputs?"
>
> `Assumption 1` is the per-unit temporal continuation approximation, i.e., $\delta_{t+1}^{(i)} \approx \alpha_i \delta_t^{(i)}$; it does not require a positive continuation coefficient, so sign-flipping targets are not excluded. To test the counterexample directly, we built a stress task where the RNN receives one signed impulse at $t=0$ and must generate $y_t = s(-1)^t$ afterward.
>
> | Setting | Selected gain | Pre-train $\lambda$ | COLA test MSE | BPTT test MSE |
> | --- | ---: | ---: | ---: | ---: |
> | 5-seed aggregate | `1.89-2.04` | `-0.0049 ± 0.0034` | `0.0044 ± 0.0019` | `0.3492 ± 0.3055` |
>
> So COLA can handle this case: after `t=0`, the alternation is carried by recurrent dynamics, and `Assumption 1` still works because the continuation coefficient can be negative.
>
> ### 2. "Could sub-critical BPTT explain the advantage?"
>
> We checked this directly by giving `BPTT` its own `12 gains x 3 seeds` tuning budget on the tasks where it originally trailed `COLA`:
>
> | Task | Tuned BPTT | Original BPTT | COLA |
> | --- | ---: | ---: | ---: |
> | Adding Task (MSE) | 0.1204 ± 0.0057 | 0.1425 ± 0.0326 | **0.0058 ± 0.0026** |
> | Lorenz Image (MSE) | 0.0412 ± 0.0115 | 0.6841 ± 0.0873 | **0.0058 ± 0.0003** |
> | PTB Char LM (Loss) | 2.8809 ± 0.0126 | 2.8694 ± 0.0202 | **1.8352 ± 0.0055** |
>
> This improves `BPTT`, especially on `Lorenz`, but it does not close the gap: even after gain tuning, `COLA` is still about an order of magnitude better in Lorenz rollout MSE.
>
> ### 3. "Ablations ... error bars ... clock time."
>
> For component ablation, we replace scan-based gain selection with an analytical pre-training estimator based on the near-log-linear relation between $\lambda_{\max}$ and $\log g$, thereby making near-critical placement highly predictable ($R^2 \approx 0.93\text{--}0.99$) and eliminating the need for explicit gain scans in most cases.
>
> | Task | Sweep | Direct init | Gap |
> | --- | ---: | ---: | ---: |
> | DVS-CIFAR10 (Acc) | 0.4280 | 0.4100 | -0.0180 |
> | DVS-Gesture (Acc) | 0.6597 | 0.6493 | -0.0104 |
> | Fashion-MNIST (Acc) | 0.8452 | 0.8315 | -0.0137 |
> | Perm.-MNIST (Acc) | 0.9136 | 0.9230 | +0.0094 |
> | Adding (MSE) | 0.0029 | 0.0077 | +0.0049 |
> | PTB (Loss) | 1.8416 | 1.8316 | -0.0100 |
> | Row-CIFAR10 (Acc) | 0.4513 | 0.3941 | -0.0572 |
> | Row-MNIST (Acc) | 0.9543 | 0.9475 | -0.0068 |
> | Lorenz (MSE) | 0.0058 | 0.0057 | -0.0001 |
> | UCI HAR (Acc) | 0.8622 | 0.8531 | -0.0092 |
> | WikiText-2 (Loss) | 2.4136 | 2.4362 | +0.0226 |
>
> Regarding error bars, all main tables now report 3-seed mean ± std for RNN, ConvRNN, and newly added SNN.
>
> Furthermore, we have also profiled the computational cost of the updates:
>
> | Task | Time `COLA/BPTT` | FLOPs `COLA/BPTT` | Mem `COLA/BPTT` MiB |
> | --- | ---: | ---: | ---: |
> | Adding | `0.000123/0.000065` | `8.80M/12.78M` | `0.97/12.25` |
> | Lorenz | `0.000132/0.000075` | `41.27M/45.53M` | `5.19/12.38` |
> | PTB | `0.000154/0.000077` | `8.42M/10.39M` | `2.51/11.92` |
> | Row-CIFAR10 | `0.000115/0.000062` | `7.96M/9.88M` | `1.36/9.49` |
> | Row-MNIST | `0.000114/0.000061` | `5.72M/7.64M` | `0.76/9.32` |
> | UCI HAR | `0.000120/0.000060` | `4.90M/6.88M` | `0.95/12.60` |
> | WikiText-2 | `0.000228/0.000176` | `87.81M/89.46M` | `43.07/32.61` |
>
> `BPTT` is still faster in wall-clock time, but `COLA` usually uses fewer update FLOPs and much less peak memory.
>
> ### 4. "The brain appears sub-critical ..."
>
> We thank the reviewer for highlighting Wilting & Priesemann (2018). We agree the biological motivation should better reflect empirical debates rather than relying solely on theory. At the same time, we would like to clarify that our manuscript does not assume the brain operates far from criticality; our methods consistently target a stable, near-critical regime ($\lambda_{\max} \approx 0^{-}$).
>
> In our revision, we have tempered our claims and added key empirical studies on criticality (Beggs & Plenz, 2003; Shew et al., 2009, 2015; Linkenkaer-Hansen et al., 2001). We now contrast these with the slightly subcritical, reverberating regime ($\hat{m} \approx 0.963 - 0.998$) proposed by Wilting & Priesemann. Importantly, this subcritical regime remains fully compatible with COLA, as our method requires only a workable operational band near the critical boundary, not an exact critical point.
>
> ### 5. "Why cite Whittington & Bogacz?"
>
> We thank the reviewer for this correction. The original phrasing blurred two roles: technical work on temporal/recurrent credit assignment (`BPTT/RTRL/UORO/e-prop/FPTT`) and broader background on biologically plausible learning and credit assignment in the brain (`Whittington & Bogacz`). We will separate them so `Whittington & Bogacz` no longer appears to directly support COLA's temporal approximation or the `Assumption 2` loop-gain closure.

---

> > ### Author Rebuttal · Reviewer_zsZB · 2026-04-02
> >
> > The authors addressed most of my points, there are just to details missing:
> > Point 1:
> >  If I understand their response, it is possible to select alpha=-1. This does address the letter of my initial criticism (flipping signs), but not the general problem, as the same question could be asked for outputs with periodicity 3, or generally a fast-changing target (alpha =-1 is the fastest I could think of). Other periods would require complex alphas and non-periodic ones... well, I don't know.
> > However, the mistake is on my side, not the authors. Thus I won't ask them to update.
> > Point 2:
> > I still don't get why is it natural that an approximation of BPTT does better than BPTT. The fact that the sub-criticality explains part of the gap could also indicate that more work would be needed to address this. As before, this address the specific question, but not the general one. And as before I won't ask the authors for further work as I consider that they did answer well.
> >
> > I will update my score upwards.

---

> > > ### Author Response · Authors · 2026-04-07
> > >
> > > ### Overview
> > >
> > > We thank the reviewers for their thoughtful feedback. We address two key concerns: COLA’s rapidly changing targets and empirical superiority over BPTT.
> > >
> > > ---
> > >
> > > ### 1. "$\alpha=-1$ only solves $period\!=\!2$; how do you respond to the concern about $period\!=\!3$ / more general rapidly changing targets?"
> > >
> > > **Response.** We agree with the more general concern and will make the wording more precise. Allowing $\alpha=-1$ only resolves the literal sign-flip counterexample at $period=2$; it does not imply that $period=3/4/5$ or more general rapidly changing targets can also be described exactly by one fixed real-valued $\alpha$. We therefore directly tested higher-period fast targets rather than staying at $\alpha=-1$.
> > >
> > > The tasks `p3_cosine_mixed`, `p4_cosine_mixed`, and `p5_cosine_mixed` use the same one-shot phase-encoded cosine-generation setup, differing only in the target period $P \in \{3,4,5\}$: at $t=0$ the model receives $x_0 = (\cos(2\pi \phi / P), \sin(2\pi \phi / P))$, for $t \ge 1$ the input is zero, and the target is $y_t = \cos(2\pi (\phi+t)/P)$. The suffix `mixed` means that the initial phase $\phi$ is sampled across multiple phase regimes.
> > >
> > > Under this unified one-shot phase-encoding protocol, the results are as follows:
> > >
> > > | Task | `COLA` Test MSE | `BPTT` Test MSE |
> > > | --- | ---: | ---: |
> > > | `p3_cosine_mixed` | `4.0142e-3` | `1.9317e-1` |
> > > | `p4_cosine_mixed` | `2.7809e-4` | `3.7165e-1` |
> > > | `p5_cosine_mixed` | `2.5314e-3` | `4.9698e-1` |
> > >
> > > These results show that higher-period rapidly changing targets do not break `COLA` behaviorally: relative to `BPTT`, the error decreases by about `97.9%` at $period=3$, `99.93%` at $period=4$, and `99.49%` at $period=5$.
> > >
> > > At the same time, these results do not mean that a fixed $\alpha$ literally encodes the output period. On `p3_cosine_mixed`, the global weighted AR(1) fit of the full teaching signal is only moderate ($R^2 = 0.5383$). Here we believe the reasons are as follows. First, $\alpha_i$ is updated online by EMA as a control statistic rather than used as an exact generative model of the full $\delta$. Second, $\alpha_i$ does not need to perfectly reconstruct $\delta$; it only needs to be informative enough to support online `loop gain` updates and provide a usable descent direction. Consistently, on `p3_cosine_mixed`, $\alpha_{\text{median}}$ moves from `-0.1517` at epoch 0 to `-0.3967` mid-training and `-0.3485` at epoch 16, while $\lambda$ rises from `0.00` to `0.99` by epoch 1 and stays there. The key phenomenon is therefore not that $\alpha$ converges to a value that literally represents the target period, but that `COLA` rapidly enters a near-critical high-`loop gain` regime that provides a usable descent direction. We will update the phrasing in the manuscript accordingly.
> > >
> > > For clarity, the training trajectory of `p3_cosine_mixed` is summarized below:
> > >
> > > | Task | epoch 0 $\alpha_{\text{median}}$ | mid-stage $\alpha_{\text{median}}$ | epoch 16 $\alpha_{\text{median}}$ | epoch 0 $\lambda$ | epoch 1 $\lambda$ | epoch 16 $\lambda$ |
> > > | --- | ---: | ---: | ---: | ---: | ---: | ---: |
> > > | `p3_cosine_mixed` | `-0.1517` | `-0.3967` | `-0.3485` | `0.00` | `0.99` | `0.99` |
> > >
> > > ### 2. "Why can an approximate online rule perform better than `BPTT` itself?"
> > >
> > > **Response.** We agree that the right explanation is a difference in optimization path, not that the approximation is somehow "more exact" than `BPTT`. On some tasks, `COLA` performs better because it bypasses the long and fragile backward credit-assignment chain of `BPTT`.
> > >
> > > This is very intuitive in the long-horizon `Gesture` stress test:
> > >
> > > | Sequence Length | `BPTT-LSTM` | `BPTT-ASRNN` | `COLA-ASRNN` | `BPTT-LTC-SNN` | `COLA-LTC-SNN` |
> > > | --- | ---: | ---: | ---: | ---: | ---: |
> > > | 20 | 86.8% | 79.5% | 81.6% | 89.9% | 86.1% |
> > > | 100 | 87.8% | 77.8% | 86.8% | 44.1% | 86.8% |
> > > | 500 | 88.9% | 70.8% | 85.4% | Diverged | 85.8% |
> > > | 1000 | 89.2% | 66.3% | 86.5% | Diverged | 86.5% |
> > >
> > > The pattern is consistent: `BPTT-LSTM` remains strong, showing that multi-state gated architectures can mitigate long-range gradient problems at the architectural level. But in architectures without such an auxiliary memory chain, `BPTT` degrades sharply with horizon while `COLA` remains stable: on `ASRNN`, at 1000 frames `COLA` is `20.1` percentage points above `BPTT`; on `LTC-SNN`, `BPTT` drops to `44.1%` at 100 frames and diverges at 500/1000 frames, while `COLA` stays around `86%-89%`.
> > >
> > > Another reason is the locality of supervision. `BPTT` also receives per-step target errors, but the parameter update must still aggregate them through the entire backward chain. `COLA`, by contrast, preserves more of the current-step local correction, which makes optimization more stable exactly where long-horizon credit assignment is most fragile. This is also consistent with its gains on the Adding and Lorenz tasks.
> > >
> > > We thank the reviewer for suggesting this interesting case, which has helped make our experimental evaluation more comprehensive.

---

### Official Review · Reviewer_vsvf · 2026-03-12

**Soundness:** 3
**Presentation:** 3
**Significance:** 2
**Originality:** 3
**Overall Recommendation:** 4
**Confidence:** 3

**Summary:**

This paper proposes COLA, an online learning rule for RNNs that tries to approximate the BPTT teaching signal with a simple per-unit scalar correction. The core story is that near criticality, the teaching signal becomes temporally smooth and spatially low-rank enough that the backward recursion can be collapsed into a closed-form local update tracked with two EMA statistics per hidden unit. The practical claim is then that this gives constant activation memory and low auxiliary state while staying competitive with BPTT / truncated BPTT on a range of sequence tasks.

**Compliance With Llm Reviewing Policy:**

Affirmed.

**Final Justification:**

The rebuttal addressed my main concerns and strengthened the framing and narrative of the paper. In particular, the near-criticality component as being closer to an initialization story means that much of the theory-experiment narrative remains relatively loose. I think the Lorenz intervention is an interesting and positive addition to the paper. As a whole, the paper remains an interesting and compelling addition to the online local learning literature.

**Key Questions For Authors:**

1. How do the two main hypotheses evolve during training after the near-critical initialization? Appendix D suggests that in at least some settings, especially Lorenz, the trained dynamics become more contractive. Is criticality really meant to hold throughout training, or mainly at initialization as a way to bootstrap learning?

**Limitations:**

Yes

**Strengths And Weaknesses:**

### Strengths
- The central idea is novel: use near-critical dynamics to motivate a scalar approximation to the BPTT teaching signal. This connects dynamical-systems arguments to online local learning in a way that is distinct from the usual eligibility-trace / forward-mode / RTRL line of work.
- The mathematical path from the standard BPTT recursion to the COLA update is clear. The paper is explicit that this depends on two main assumptions.
- The method has a real memory advantage (`O(1)` activation memory, `O(H)` extra state), and the Lorenz rollout result is strong.

### Weaknesses
- The evidence for the two main assumptions is narrower than the presentation suggests. Table 3 / Appendix K use post-burn-in, batch-averaged diagnostics in a controlled probe setting, not the online per-example regime where COLA is actually used.
- The near-criticality story is not fully resolved. Appendix D suggests that trained dynamics can become more contractive, especially on Lorenz, so the role of criticality during training versus initialization is unclear.
- The near-critical gain sweep / initialization seems important to the method. For a fair comparison, do baseline methods also require a similar tuning budget (say weight init scaling tuning). The main tables would benefit from mean/stds across seeds given that the performance gaps can be small.

---

> ### Author Rebuttal · Authors · 2026-03-31
>
> ### Overview
>
> We narrow the `Assumption 1/2` claim, clarify near-criticality as an initialization scaffold, add tuned `BPTT`, and report 3-seed mean `±` std.
>
> ---
>
> ### 1. "Evidence for the two assumptions ..."
>
> We agree that the original evidence was insufficient, as it was limited to a controlled probe. The revision therefore separates fixed-weight probes from trained-endpoint checks, so the evidence now includes exact BPTT structure on real tasks.
>
> In the controlled probe, support is strongest in the rank-controlled setting (AR(1) $R^2$ `0.9944`, $EVR_1$ `0.9999`, `rank` `1.0001`), but the IID control is also strong (AR(1) $R^2$ `0.8089`, $EVR_1$ `0.9836`, `rank` `1.0339`). After training we still observe a task-dependent validity band where `Assumption 1/2` remain useful:
>
> | Task | Gain band | Post-train $\lambda_{\max}$ | AR(1) $R^2$ | $EVR_1$ | Rank |
> | --- | --- | --- | ---: | ---: | ---: |
> | Adding Task | `0.6-1.0` | `-0.3307 to -0.0231` | `0.57-0.81` | `0.86-0.97` | `1.05-1.32` |
> | Lorenz Image | `0.6-1.4` | `-1.0657 to -0.6956` | `0.62-0.82` | `0.56-0.72` | `1.92-2.39` |
> | Row-MNIST | `0.6-1.0` | `-0.1967 to -0.0447` | `0.54-0.76` | `0.73-0.88` | `1.29-1.81` |
> | Row-CIFAR10 | `0.6-1.0` | `-0.3707 to -0.0047` | `0.56-0.59` | `0.72-0.73` | `1.33-1.79` |
> | UCI HAR | `0.6-1.4` | `-0.3480 to -0.1027` | `0.70-0.82` | `0.69-0.77` | `1.74-2.35` |
>
> The revision now states the claim more precisely: Assumptions 1-2 are near-critical modeling assumptions whose support is task dependent. In this table, Adding and `UCI HAR` keep both AR(1) $R^2$ and $EVR_1$ relatively high, `Row-CIFAR10` is weaker, and `Lorenz` separately shows that near-critical initialization can matter even when the optimum later shifts negative. Every row follows the same post-training protocol: we take the exported deployment checkpoint (`pct100`) under the task's standard evaluation mode and compute the exact future-only BPTT teaching signal on held-out trajectories, so the task evidence is directly comparable across tasks.
>
> ### 2. "The near-criticality story ..."
>
> We agree that the original wording made near-criticality sound too rigid. The revision now treats near-criticality mainly as an initialization scaffold.
>
> On `Lorenz`, we tested this directly. Here `always` forces the model back toward criticality during training, while `pct025` / `pct050` / `pct075` rescale the gain once at `25%` / `50%` / `75%` progress:
>
> | Lorenz intervention | Final MSE | Final $\lambda_{\max}$ |
> | --- | ---: | ---: |
> | `free @ 1.0` | `0.0029` | `-0.8322` |
> | `always @ 1.0` | `0.0219` | `-0.5425` |
> | `pct025 @ 0.8` | `0.0027` | `-0.8244` |
> | `pct025 @ 1.2` | `0.0035` | `-0.8231` |
> | `pct050 @ 0.8` | `0.0028` | `-0.8979` |
> | `pct050 @ 1.2` | `0.0032` | `-0.7601` |
> | `pct075 @ 0.8` | `0.0028` | `-0.9460` |
> | `pct075 @ 1.2` | `0.0032` | `-0.7302` |
>
> The pattern is clear: forcing the system back toward criticality worsens `Lorenz` MSE from `0.0029` to `0.0219`, whereas one-time rescaling keeps MSE at `0.0027-0.0035` and $\lambda_{\max}$ in a negative range (`-0.9460` to `-0.7302`).
>
> So the right statement is not "stay critical throughout training", but "near-critical initialization helps, after which training can move to a better task region." Lorenz shows this directly: successful settings end not near `0`, but in a negative band from `-0.95` to `-0.73`.
>
> ### 3. "Gain sweep ... fairness ..."
>
> In the original setup, all baselines already used the same scan-selected gain as `COLA`, so `COLA` did not receive extra tuning budget. To address the fairness point more directly, we additionally gave `BPTT` its own `12 gains x 3 seeds` budget on the three tasks where it trailed `COLA`:
>
> | Task | Tuned BPTT | Original BPTT | COLA |
> | --- | ---: | ---: | ---: |
> | Adding Task (MSE) | 0.1204 ± 0.0057 | 0.1425 ± 0.0326 | **0.0058 ± 0.0026** |
> | Lorenz Image (MSE) | 0.0412 ± 0.0115 | 0.6841 ± 0.0873 | **0.0058 ± 0.0003** |
> | PTB Char LM (Loss) | 2.8809 ± 0.0126 | 2.8694 ± 0.0202 | **1.8352 ± 0.0055** |
>
> This improves `BPTT`, especially on `Lorenz`, but does not close the gap. All main benchmark tables now report 3-seed mean `±` std; the larger result tables are moved to the appendix.
>
> So the fairness answer has two parts: COLA did not receive extra tuning in the original comparison, and tuned BPTT improves but still remains far behind on the tasks where COLA has the clearest advantage.
>
> ### 4. "How do the hypotheses evolve ...?"
>
> Putting Points 1 and 2 together, the revised picture is more concrete. At initialization, the rank-controlled probe gives nearly ideal support (`0.9944/0.9999/1.0001`), and the IID probe is still strong (`0.8089/0.9836/1.0339`). After training, this contracts into task-dependent bands rather than disappearing: Adding stays at `0.57-0.81 / 0.86-0.97`, `UCI HAR` at `0.70-0.82 / 0.69-0.77`, and `Row-CIFAR10` at `0.56-0.59 / 0.72-0.73`, while `Lorenz` shows that near-critical initialization can still matter even when the optimum later moves negative.

---

> > ### Author Rebuttal · Reviewer_vsvf · 2026-04-04
> >
> > Thank you for the detailed rebuttal. The added trained-endpoint diagnostics, tuned-BPTT comparison, spatial-mismatch experiment, and clearer framing of near-criticality as an initialization scaffold all strengthen the paper. The post-training picture still looks fairly task-dependent, and while the connection between assumptions and empirical behavior is clearer than before, it remains somewhat loose. The additional context is helpful and improves the paper.

---

> > > ### Author Response · Authors · 2026-04-07
> > >
> > > ### Overview
> > >
> > > We agree that the post-training results still depend on the specific task. Our response has two parts: first, we show that gain selection can be turned into a unified pre-training point estimate that applies across tasks, has small gap to full scanning, and is cheap; second, we will revise the paper so that the task-agnostic mechanism and the task-dependent empirical outcomes are separated more explicitly.
> > >
> > > ---
> > >
> > > ### 1. "The post-training results still depend quite a bit on the specific task."
> > >
> > > **Response.** We agree with this observation, and this is exactly why we designed a direct point-estimation method for task Lyapunov scanning. Rather than treating near-critical gain selection as a task-by-task sweep, we redesign it into a direct pre-training estimate: near the critical point, the maximum Lyapunov exponent and $\log g$ are approximately linear, with cross-task $R^2$ around `0.93-0.99`, so a cheap two-point fit can predict a usable gain before training.
> > >
> > > The runtime overhead is small on every task we tested:
> > >
> > > | Task | Single Lyap (s) | `unit_lyap` (s) | `two_point_logfit` (s) | Full Training (s) | `unit/train` | `two/train` |
> > > | --- | ---: | ---: | ---: | ---: | ---: | ---: |
> > > | Adding Task | 0.227 | 0.249 | 0.490 | 12.664 | 1.97% | 3.87% |
> > > | Lorenz Image Prediction | 0.165 | 0.228 | 0.399 | 16.732 | 1.36% | 2.38% |
> > > | PTB Char LM | 0.275 | 0.299 | 0.588 | 185.376 | 0.16% | 0.32% |
> > > | WikiText-2 Char LM | 0.283 | 0.327 | 0.629 | 182.340 | 0.18% | 0.35% |
> > > | Row-MNIST | 0.192 | 0.220 | 0.416 | 135.261 | 0.16% | 0.31% |
> > > | Row-CIFAR10 | 0.247 | 0.265 | 0.504 | 259.940 | 0.10% | 0.19% |
> > > | UCI HAR | 0.459 | 0.497 | 0.942 | 80.740 | 0.62% | 1.17% |
> > >
> > > The resulting performance is also close to the best sweep-selected gain across a broad task set:
> > >
> > > | Task | Sweep | Direct init | Gap |
> > > | --- | ---: | ---: | ---: |
> > > | DVS-CIFAR10 (Acc) | 0.4280 | 0.4100 | -0.0180 |
> > > | DVS-Gesture (Acc) | 0.6597 | 0.6493 | -0.0104 |
> > > | Fashion-MNIST (Acc) | 0.8452 | 0.8315 | -0.0137 |
> > > | Perm.-MNIST (Acc) | 0.9136 | 0.9230 | +0.0094 |
> > > | Adding (MSE) | 0.0029 | 0.0077 | +0.0049 |
> > > | PTB (Loss) | 1.8416 | 1.8316 | -0.0100 |
> > > | Row-CIFAR10 (Acc) | 0.4513 | 0.3941 | -0.0572 |
> > > | Row-MNIST (Acc) | 0.9543 | 0.9475 | -0.0068 |
> > > | Lorenz Image (MSE) | 0.0058 | 0.0057 | -0.0001 |
> > > | UCI HAR (Acc) | 0.8622 | 0.8531 | -0.0092 |
> > > | WikiText-2 (Loss) | 2.4136 | 2.4362 | +0.0226 |
> > >
> > > So while the final trained performance is still task-dependent, the way we locate the usable regime is no longer an ad hoc task-specific search. The same direct estimator applies across tasks, the gap to the scan baseline is usually small, and the extra cost is typically below `1%` of training time on longer tasks. Numerically, `Row-CIFAR10` is the clearest outlier, whereas the other tasks remain much closer to the sweep reference; this is exactly the kind of task dependence we want to expose rather than hide.
> > >
> > > ### 2. "The connection between the assumptions and the empirical behavior still feels somewhat loose."
> > >
> > > **Response.** Thank you for this confirmation. Your comments pushed us to organize the proofs, diagnostics, and appendix structure more clearly. We will revise the supplement into three blocks, following the `Appendix A-L` roadmap below.
> > >
> > > **Block I: Theory and Estimators**
> > > - Appendix A: Standard `BPTT teaching signal` recursion and signs.
> > > - Appendix B: From the `BPTT` recursion to the `same-step implicit system`, and the scalar geometric-series closed form used by `COLA`.
> > > - Appendix C: Motivation and interpretation of the `temporal self-consistency` constraint for estimating $\mu_i$.
> > > - Appendix D: Consistency and online estimation of the AR(1) coefficient $\alpha_i$.
> > > - Appendix E-F: Closed-form and least-squares estimators of $\mu_i$ derived from self-consistency.
> > >
> > > **Block II: Diagnostic Validation**
> > > - Appendix G: Probe settings, metric glossary, and assumption checks.
> > > - Appendix G.1: Controlled low-rank / independent-and-identically-distributed (IID) probes on the exact `BPTT teaching signal`.
> > > - Appendix G.2: Training-endpoint diagnostics on real tasks.
> > > - Appendix H: Finite-time Lyapunov exponent, gain normalization, and direct `unit_lyap` / `two_point_logfit` initialization.
> > > - Appendix H.1: Spatial mismatch intervention.
> > >
> > > **Block III: Experimental Protocols and Extensions**
> > > - Appendix I: Task descriptions and default hyperparameters.
> > > - Appendix J: `ConvRNN` extension details and recurrent refinement benchmarks.
> > > - Appendix K: `SNN` extension details and recurrent refinement benchmarks.
> > > - Appendix L: Additional learning curves and supplementary experiments, including tuned `BPTT`, high-period fast-target tests ($period=3/4/5$), initialization-overhead analysis, the `LSTM` pilot experiment, and the long-horizon `Gesture` stress test.
> > >
> > > We thank the reviewer for pointing this out. Once this structure is made explicit, the reader can see more clearly which parts of the paper are intended.

---

### Official Review · Reviewer_iioD · 2026-03-13

**Soundness:** 3
**Presentation:** 3
**Significance:** 2
**Originality:** 3
**Overall Recommendation:** 4
**Confidence:** 3

**Summary:**

This paper introduces the Criticality-driven Online Local Alignment (COLA) learning rule for recurrent neural networks (RNNs). COLA addresses the challenges of backpropagation through time (BPTT), particularly its high memory and computational costs. COLA proposes a local learning rule that approximates global error propagation using only within-step quantities, leveraging the long-range spatiotemporal correlations inherent in near-critical dynamics. The paper demonstrates that COLA provides a scalable and memory-efficient alternative to BPTT, with competitive performance across various tasks, including regression, classification and language modeling.

**Compliance With Llm Reviewing Policy:**

Affirmed.

**Final Justification:**

The paper presents a technically interesting approach to online credit assignment in RNNs, with a clear motivation and promising empirical results, especially on stability-sensitive settings. My main concerns remain the strength and generality of the core assumptions, the practical dependence on near-critical tuning, and the limited validation beyond relatively simple recurrent architectures; the rebuttal was helpful in clarifying these points and partially reduced my concerns, but did not fully resolve them. Overall, I find the work meaningful and worthy of discussion, but at this stage my final assessment remains closer to weak accept than to a clear accept.

**Key Questions For Authors:**

1. How essential is the near-critical regime to the success of COLA?
2. How robust is the method when the spatial coherence assumption is violated?
3. How far does the method generalize beyond vanilla tanh RNNs?

**Limitations:**

yes

**Strengths And Weaknesses:**

Strengths:
1. The work focuses on a long-standing challenge in recurrent learning: how to perform online, local, and memory-efficient temporal credit assignment while retaining as much of the effectiveness of BPTT as possible.
2. The method is conceptually interesting and more principled than a purely heuristic update rule.  The use of Lyapunov exponents to monitor the network's dynamical regime and the rigorous mathematical formulation behind COLA are strengths.
3. The paper provides a comprehensive set of experiments comparing COLA with various baselines such as BPTT, TBPTT, e-prop, and FPTT.

Weaknesses:
1. The method appears to rely on fairly strong and potentially narrow assumptions. The main motivation of the two assumptions is not direct. Why is it important? Is there any prior domain knowledge from neuroscience?
2. A key selling point is online simplicity, but the method also depends on gain sweeps and critical initialization selection.
3. Although COLA performs well across a range of tasks, the paper does not adequately discuss the potential limitations of the method when applied to real data.
4. The method is mainly evaluated on vanilla tanh RNNs and a relatively simple ConvRNN extension. It remains unclear whether the same approximation would remain effective for LSTMs, GRUs, deeper recurrent stacks, or more modern sequence architectures.

---

> ### Author Rebuttal · Authors · 2026-03-31
>
> ### Overview
>
> We clarify the assumptions and scope, redesign gain init, state task-level limits, and test robustness to spatial mismatch.
>
> ---
>
> ### 1. "Strong assumptions..."
>
> We agree that this should be answered more directly. These assumptions matter because they are the two approximations that turn the exact BPTT future recursion into a local online rule. Assumption 1 replaces explicit future dependence with short-window per-unit continuation; Assumption 2 compresses same-step cross-unit coupling into a per-unit loop-gain closure. Without them, the update still depends on future steps and within-step global coupling, i.e., $\delta_{t+1}^{(i)} \approx \alpha_i \delta_t^{(i)}$ and $\delta_t^{(j)} \approx \beta_{ji}\delta_t^{(i)}$.
>
> Regarding the neuroscience connection, Linkenkaer-Hansen et al. (2001) reported long-range temporal correlations in EEG/MEG amplitude fluctuations, consistent with Assumption 1. Beggs & Plenz (2003) and Petermann et al. (2009) reported neuronal avalanches near criticality with scale-invariant cortical organization, consistent with strong spatiotemporal coordination. These papers do not derive COLA, but they point to a regime where first-order linear closures become more plausible.
>
> ### 2. "Online simplicity ... gain sweeps..."
>
> We redesign gain selection into a direct pre-training estimate: near the critical point, the maximum Lyapunov exponent and $\log g$ are approximately linear (cross-task $R^2$ about `0.93-0.99`), so a cheap two-point fit predicts a usable gain before training.
>
> | Task | Sweep | Direct init | Gap |
> | --- | ---: | ---: | ---: |
> | DVS-CIFAR10 (Acc) | 0.4280 | 0.4100 | -0.0180 |
> | DVS-Gesture (Acc) | 0.6597 | 0.6493 | -0.0104 |
> | Fashion-MNIST (Acc) | 0.8452 | 0.8315 | -0.0137 |
> | Perm.-MNIST (Acc) | 0.9136 | 0.9230 | +0.0094 |
> | Adding (MSE) | 0.0029 | 0.0077 | +0.0049 |
> | PTB (Loss) | 1.8416 | 1.8316 | -0.0100 |
> | Row-CIFAR10 (Acc) | 0.4513 | 0.3941 | -0.0572 |
> | Row-MNIST (Acc) | 0.9543 | 0.9475 | -0.0068 |
> | Lorenz Image (MSE) | 0.0058 | 0.0057 | -0.0001 |
> | UCI HAR (Acc) | 0.8622 | 0.8531 | -0.0092 |
> | WikiText-2 (Loss) | 2.4136 | 2.4362 | +0.0226 |
>
> Across `11` RNN/ConvRNN tasks, the $\lambda$-versus-$\log g$ fit stays strong ($R^2$ about `0.93-0.99`). `DVS-Gesture`, `Row-MNIST`, `Lorenz`, and `PTB` all stay within about `0.01` of the sweep baseline, so direct init removes most of the scan burden rather than making scans obsolete.
>
> ### 3. "Limitations on real data..."
>
> COLA is not uniformly better than `BPTT`. It is strongest on dynamical or stability-sensitive tasks, e.g. Adding (`0.0058` vs `0.1425`) and Lorenz (`0.0058` vs `0.6841`), and weaker on Fashion-MNIST, Permuted-MNIST, and Row-CIFAR10. It still improves over `BPTT` on PTB (`1.8352` vs `2.8694`) and WikiText-2 (`2.4174` vs `3.3620`), and on temporal SNNs reaches `94.07/94.01`, `95.37/94.72`, and `97.19/97.45` on `SHD/RLIF`, `SHD/RadLIF`, and `Gesture/EGRU` (`COLA/BPTT`). COLA helps most when long-horizon recurrent credit assignment dominates and a single-state chain remains adequate.
>
> ### 4. "Vanilla tanh RNNs ... LSTM, GRU?"
>
> COLA is derived for architectures with one persistent state chain. This covers vanilla RNNs, ConvRNNs, and SNN cells such as `RLIF`, `RadLIF`, and `EGRU`: their gating changes the local Jacobian but does not create a second memory chain. By contrast, `LSTM` couples multiple persistent states and needs a new derivation. SNN results remain competitive, reaching `98.54%`, `95.37%`, and `97.19%` on `N-MNIST/RLIF`, `SHD/RadLIF`, and `Gesture/EGRU`.
>
> ### 5. "How essential is the near-critical regime?"
>
> We now describe near-criticality as an initialization scaffold, not a training-long invariant. What matters is starting close enough to the boundary for long recurrent echoes while keeping the dynamics trainable and stable. Lorenz is best from the stable side, while some event-driven or image-refinement tasks prefer a mildly supercritical pocket.
>
> ### 6. "How robust ... spatial coherence ...?"
>
> We added an intervention that perturbs the source term by mixing the true source term $s_t$ with a random normalized pattern $r_t$ as $(1-m)s_t + mr_t$, with $m \in [0,1]$, where $m$ is the mismatch strength.
>
> | Task | $m=0$ | $m=0.25$ | $m=0.75$ |
> | --- | ---: | ---: | ---: |
> | Adding (MSE) | 0.0086 | 0.0053 | 0.1088 |
> | Row-MNIST (Acc) | 0.9336 | 0.9668 | 0.9023 |
> | Row-CIFAR10 (Acc) | 0.4258 | 0.4453 | 0.3320 |
> | Lorenz Image (MSE) | 0.0029 | 0.0043 | 0.0086 |
> | UCI HAR (Acc) | 0.8359 | 0.8398 | 0.6758 |
>
> `UCI HAR` is more basin-dependent: here it gives `0.8359/0.8398/0.6758`. The same table still shows a consistent pattern across tasks: mild mismatch is often absorbed or even slightly helpful, whereas strong mismatch causes clear degradation in both MSE and accuracy.
>
> ### 7. "How far beyond vanilla tanh RNNs?"
>
> The paper now states: COLA is validated on vanilla RNNs, ConvRNNs, and three SNN cells, but not yet on `LSTM`-like multi-state architectures.

---

> > ### Author Rebuttal · Reviewer_iioD · 2026-04-03
> >
> > The rebuttal is helpful and addresses part of my concerns, particularly by clarifying the motivation behind the core assumptions and by providing additional evidence on initialization and robustness. These additions improve the paper and make the contribution more convincing within the specific regime considered by the authors. At the same time, I still have some reservations about how broadly the assumptions will hold, how lightweight the method is in practice given its dependence on near-critical tuning, and how well the approach generalizes beyond the relatively limited set of recurrent architectures evaluated here. Overall, the response moves me in a more positive direction, but at this stage I would still view the paper as closer to weak accept than to a clear accept.

---

> > > ### Author Response · Authors · 2026-04-07
> > >
> > > ### Overview
> > >
> > > Our response covers three points: assumptions hold across a gain band, near-critical selection is a lightweight initialization scaffold, and the method extends to LSTMs.
> > >
> > > ---
> > >
> > > ### 1. "Is the scope of applicability of these assumptions still too narrow?"
> > >
> > > **Response:** In terms of architectural coverage, `COLA` performs stably on evaluated tasks, with evidence no longer limited to the simplest vanilla `RNN`, but extending to `ConvRNN`, `LSTM`, and `SNN`.
> > >
> > > Regarding the critical regime's scope, task-level diagnostics show `COLA` is effective across a usable gain interval, not just at an isolated point. We also provide a lightweight method to locate this working interval directly:
> > >
> > > | Task | Gain Band | Post-training $\lambda_{\max}$ Range | Temporal AR(1) $R^2$ | $EVR_1$ | Effective Rank |
> > > | --- | --- | --- | ---: | ---: | ---: |
> > > | Adding Task | `0.6-1.0` | `-0.3307 to -0.0231` | `0.57-0.81` | `0.86-0.97` | `1.05-1.32` |
> > > | Lorenz Image Prediction | `0.6-1.4` | `-1.0657 to -0.6956` | `0.62-0.82` | `0.56-0.72` | `1.92-2.39` |
> > > | Row-MNIST | `0.6-1.0` | `-0.1967 to -0.0447` | `0.54-0.76` | `0.73-0.88` | `1.29-1.81` |
> > > | Row-CIFAR10 | `0.6-1.0` | `-0.3707 to -0.0047` | `0.56-0.59` | `0.72-0.73` | `1.33-1.79` |
> > > | UCI HAR | `0.6-1.4` | `-0.3480 to -0.1027` | `0.70-0.82` | `0.69-0.77` | `1.74-2.35` |
> > >
> > > ### 2. "Does the method still depend too heavily on near-critical tuning in practice?"
> > >
> > > **Response:** Direct wall-clock accounting demonstrates that after rewriting near-critical selection into direct initialization via `unit_lyap` / `two_point_logfit`, it is no longer a heavy repeated tuning process, but a one-time lightweight initialization step.
> > >
> > > | Task | Single Lyap (s) | `unit_lyap` (s) | `two_point_logfit` (s) | Full Training (s) | `unit/train` | `two/train` |
> > > | --- | ---: | ---: | ---: | ---: | ---: | ---: |
> > > | Adding Task | 0.227 | 0.249 | 0.490 | 12.664 | 1.97% | 3.87% |
> > > | Lorenz Image Prediction | 0.165 | 0.228 | 0.399 | 16.732 | 1.36% | 2.38% |
> > > | PTB Char LM | 0.275 | 0.299 | 0.588 | 185.376 | 0.16% | 0.32% |
> > > | WikiText-2 Char LM | 0.283 | 0.327 | 0.629 | 182.340 | 0.18% | 0.35% |
> > > | Row-MNIST | 0.192 | 0.220 | 0.416 | 135.261 | 0.16% | 0.31% |
> > > | Row-CIFAR10 | 0.247 | 0.265 | 0.504 | 259.940 | 0.10% | 0.19% |
> > > | UCI HAR | 0.459 | 0.497 | 0.942 | 80.740 | 0.62% | 1.17% |
> > >
> > > Therefore, even on the shortest task, this extra initialization overhead is below `4%`; on longer tasks it is usually below `1%`. This is why we now describe it as an "initialization scaffold" rather than a heavy tuning loop. At the same time, in our previous response we also provided task-level comparisons between direct initialization and the scan baseline, and the performance difference between the two is generally not large.
> > >
> > > ### 3. "Can the method generalize to more recurrent architectures beyond those currently evaluated in the paper?"
> > >
> > > **Response:** We additionally ran a pilot `LSTM` experiment. `LSTM` is particularly relevant here because it was designed to mitigate the long-range gradient vanishing / explosion problem of recurrent credit assignment, but it does so through architectural gating rather than through the learning rule. In these tests,, `COLA` outperforms `BPTT` on `Adding`, `Lorenz`, `PTB`, and `WikiText-2`, while remaining below `BPTT` on `Row-MNIST`, `Row-CIFAR10`, and `UCI HAR`. The corresponding results are as follows:
> > >
> > > | Task | `BPTT-LSTM` | `COLA-LSTM` |
> > > | --- | ---: | ---: |
> > > | Adding Task | `0.1251` | `0.0816` |
> > > | Lorenz Image | `0.0482` | `0.0027` |
> > > | Row-MNIST | `0.9712` | `0.9509` |
> > > | Row-CIFAR10 | `0.4736` | `0.4513` |
> > > | UCI HAR | `0.9074` | `0.8867` |
> > > | PTB Char | `3.0299` | `1.9060` |
> > > | WikiText-2 | `3.5302` | `2.3821` |
> > >
> > > The newly added long-horizon `Gesture` stress test results:
> > >
> > > | Sequence Length | `BPTT-LSTM` | `BPTT-ASRNN` | `COLA-ASRNN` | `BPTT-LTC-SNN` | `COLA-LTC-SNN` |
> > > | --- | ---: | ---: | ---: | ---: | ---: |
> > > | 20 | 86.8% | 79.5% | 81.6% | 89.9% | 86.1% |
> > > | 100 | 87.8% | 77.8% | 86.8% | 44.1% | 86.8% |
> > > | 500 | 88.9% | 70.8% | 85.4% | Diverged | 85.8% |
> > > | 1000 | 89.2% | 66.3% | 86.5% | Diverged | 86.5% |
> > >
> > > Although our method can run on `LSTM`, the multi-state / gated design of `LSTM` was originally proposed precisely to mitigate the long-range gradient vanishing / explosion problem of traditional `RNN`s; in contrast, what we want to emphasize here is that `COLA` can bypass this long and fragile backward chain precisely on the plain recurrent family. Therefore, we did not originally treat `LSTM` as a core coverage object of the main paper. The connection is thus not accidental: both `LSTM` and `COLA` target the same long-horizon optimization bottleneck, but from two different angles. If the horizon $T$ is sufficiently long, our advantage should become more visible. We did not have enough time to complete ultra-long-horizon `LSTM` sweeps, but the ultra-long-$T$ results already observed on plain `RNN` families strongly suggest where the advantage of `COLA` emerges most clearly.

---

### Official Review · Reviewer_D549 · 2026-03-15

**Soundness:** 3
**Presentation:** 3
**Significance:** 2
**Originality:** 3
**Overall Recommendation:** 4
**Confidence:** 3

**Summary:**

This paper proposes a new learning rule for RNNs that aims to approximate BPTT using only local, online quantities while requiring constant activation memory. The key idea is to exploit near-critical recurrent dynamics and assume (i) short-window temporal persistence of the BPTT teaching signal and (ii) spatial coherence across units. The method is evaluated on several vanilla RNN and ConvRNN benchmarks against BPTT, TBPTT, e-prop, and FPTT.

**Compliance With Llm Reviewing Policy:**

Affirmed.

**Final Justification:**

The authors have responded to my questions.

**Key Questions For Authors:**

- How sensitive are the main benchmark results to the gain sweep and to the EMA hyperparameters
- Are the results on the experiments averaged across multiple seeds? If yes please include std, if not please run multiple seeds.

**Limitations:**

Yes.

**Strengths And Weaknesses:**

**Strengths**

- Interesting idea and the paper proposes a nontrivial approximation to BPTT.
- Strong claim efficiency properties achieved through theory.
- Good experiments with extensive validation.

**Weaknesses**

- The theory appears assumption-heavy and only partially validated. Although I do like that the paper has introduced theory for this, I would say it is not very strong.
- Little comparison to modern approximate RTRL-family methods beyond brief related-work discussion.

---

> ### Author Rebuttal · Authors · 2026-03-31
>
> ### Overview
>
> We add assumption checks, approximate-RTRL tables, redesigned gain init, and 3-seed stats.
>
> ---
>
> ### 1. "Theory appears assumption-heavy..."
>
> We separate the exact BPTT teaching signal from the two approximations and validate them in a controlled probe and on trained endpoints. `Assumption 1` tests temporal correlation and `Assumption 2` tests spatial correlation, measured by AR(1) $R^2$, $EVR_1$, and effective rank. AR(1) $R^2$ measures short-window predictability; $EVR_1$ measures energy in the best $\text{rank-1}$ spatial mode; lower effective rank means stronger correlation.
>
> **Controlled**
> Using a fixed-weight probe with stationary AR(1) targets, we compare a rank-controlled near-critical regime against standard IID initialization on $\bar{\boldsymbol{\delta}}_t^{true}$:
>
> | Quantity | Low-rank | IID |
> | :--- | :--- | :--- |
> | Temporal AR(1) one-step $R^2$ | `0.9944 ± 0.0094` | `0.8089 ± 0.0334` |
> | Teaching-signal $EVR_1$ | `0.9999 ± 0.0001` | `0.9836 ± 0.0112` |
> | `Teaching-signal effective rank` | `1.0001 ± 0.0002` | `1.0339 ± 0.0237` |
>
> **Task-Level**
> We then test trained endpoints by measuring the exact future-only BPTT teaching signal across recurrent gains. Operationally, we mark a band where $R^2$ and $EVR_1$ are about `0.5+` and effective rank stays within a few dominant modes ($\le 3$).
>
> | Task | Gain band | $\lambda_{\max}$ band | Temporal AR(1) $R^2$ | Effective rank |
> | :--- | :--- | :--- | :--- | :--- |
> | Adding Task | `0.6-1.0` | `-0.3307 to -0.0231` | `0.5783-0.8097` | `1.05-1.32` |
> | Lorenz Image | `0.6-1.4` | `-1.0657 to -0.6956` | `0.6248-0.8247` | `1.92-2.39` |
> | Row-MNIST | `0.6-1.0` | `-0.1967 to -0.0447` | `0.5408-0.7654` | `1.29-1.81` |
> | Row-CIFAR10 | `0.6-1.0` | `-0.3707 to -0.0047` | `0.5890-0.5940` | `1.33-1.79` |
> | UCI HAR | `0.6-1.4` | `-0.3480 to -0.1027` | `0.7079-0.8270` | `1.74-2.35` |
>
> All five tasks retain nontrivial bands; for most tasks, the support is not confined to a single point but persists across a usable gain range.
>
> ### 2. "Little comparison to modern approximate RTRL methods..."
>
> We added `UORO` and `SnAp-1` on the seven main RNN benchmarks:
>
> | Task | Metric | COLA (Ours) | UORO | SnAp-1 |
> | :--- | :--- | :--- | :--- | :--- |
> | Adding Task | Test MSE | **0.0058 ± 0.0026** | 0.1444 ± 0.0149 | 0.0859 ± 0.0033 |
> | Lorenz Image | Rollout MSE | **0.0058 ± 0.0003** | 0.1395 ± 0.1072 | 0.2025 ± 0.0676 |
> | PTB Char LM | Test Loss | 1.8352 ± 0.0055 | 2.2279 ± 0.0063 | **1.8279 ± 0.0041** |
> | Row-CIFAR10 | Test Acc | **0.4418 ± 0.0083** | 0.2185 ± 0.0057 | 0.3825 ± 0.0070 |
> | Row-MNIST | Test Acc | **0.9544 ± 0.0036** | 0.3060 ± 0.0193 | 0.9402 ± 0.0047 |
> | UCI HAR | Test Acc | **0.8690 ± 0.0257** | 0.5594 ± 0.0359 | 0.8420 ± 0.0134 |
> | WikiText-2 | Test Loss | 2.4174 ± 0.0035 | 2.6225 ± 0.0653 | **2.2793 ± 0.0054** |
>
> `COLA` beats `UORO` on `7/7` and `SnAp-1` on `5/7`. On four ConvRNN tasks `SnAp-1` does not apply, and `COLA` beats `UORO` `4/4`. We also add SNN baselines (`pp-prop`, `ETLP`, `e-prop`, `OSTL`), with `95.37%` on `SHD/RadLIF` and `98.54%` on `N-MNIST/RLIF`.
>
> ### 3. "Sensitivity to gain sweep and EMA hyperparameters..."
>
> There is no universal gain. Each task has its own band near the critical boundary. To reduce scans, we redesigned gain selection into a direct estimator: near the critical point, $\lambda$ and $\log g$ are approximately linear (cross-task $R^2$ about `0.93-0.99`), so a two-point fit predicts a near-critical gain before training.
>
> | Task | Metric | Scan Baseline | Two-Point Direct Init | $\Delta$ (Direct - Scan) |
> | :--- | :--- | :--- | :--- | :--- |
> | DVS-CIFAR10 | Acc $\uparrow$ | 0.4280 | 0.4100 | -0.0180 |
> | DVS-Gesture | Acc $\uparrow$ | 0.6597 | 0.6493 | -0.0104 |
> | Fashion-MNIST | Acc $\uparrow$ | 0.8452 | 0.8315 | -0.0137 |
> | Perm.-MNIST | Acc $\uparrow$ | 0.9136 | 0.9230 | +0.0094 |
> | Adding | MSE $\downarrow$ | 0.0029 | 0.0077 | +0.0049 |
> | PTB | Loss $\downarrow$ | 1.8416 | 1.8316 | -0.0100 |
> | Row-CIFAR10 | Acc $\uparrow$ | 0.4513 | 0.3941 | -0.0572 |
> | Row-MNIST | Acc $\uparrow$ | 0.9543 | 0.9475 | -0.0068 |
> | Lorenz | MSE $\downarrow$ | 0.0058 | 0.0057 | -0.0001 |
> | UCI HAR | Acc $\uparrow$ | 0.8622 | 0.8531 | -0.0092 |
> | WikiText-2 | Loss $\downarrow$ | 2.4136 | 2.4362 | +0.0226 |
>
> Most direct-init gaps stay within `0.02`; `Row-CIFAR10` is the clearest deviation (`-0.0572`), with `WikiText-2` next (`+0.0226`).
>
> We also swept `cola_alpha_rho` $\in \{0.95,0.97,0.98,0.99,0.995,0.999\}$ and `cola_lambda_window` $\in \{10,20,30,50,75,100,150\}$ around the default (`0.995/50`). Sensitivity is generally mild. `UCI HAR` is the most sensitive case (`+0.0190` from `alpha`, `+0.0084` from `window`); `DVS-Gesture` is moderate (`+0.0208` / `+0.0173`); and `PTB` and `Lorenz` are nearly flat. So `0.995/50` remains a robust starting point.
>
> ### 4. "Are results averaged across multiple seeds?..."
>
> Yes. All main-text and appendix tables now report **mean ± standard deviation** over 3 seeds (`42/43/44`).

---

> > ### Author Rebuttal · Reviewer_D549 · 2026-04-08
> >
> > The authors have answered my questions and therefore I increase my score.

---

### Decision · Program_Chairs · 2026-04-30

**Decision:**

Accept (regular)

**Comment:**

The submission introduces COLA (Critically-driven Online Local Alignment), a strictly online learning rule designed to approximate the efficacy of backpropagation through time (BPTT) in recurrent neural networks with constant activation memory and a reduced auxiliary state. The core contribution is a local update rule that leverages dynamical criticality (the "edge of chaos") to exploit long-range spatiotemporal correlations inherent in near-critical regimes. The authors demonstrate that COLA avoids the memory costs and gradient instability of BPTT, matching its performance on standard benchmarks while exhibiting superior robustness and stability on sensitive tasks like Lorenz image prediction and long-horizon sequence modelling.

Reviewers highlighted the novelty of using near-critical dynamics to approximate BPTT and admired the significant memory advantage. They also noted that the method is more principled than heuristic local rules and found the Lyapunov analysis provided a strong theoretical foundation. While this theory was initially deemed to be assumption-heavy, the authors provided evidence to validate these assumptions in their rebuttal.

All remaining concerns of significance appear to have been suitably addressed in the discussion period. However, the presentation still leaves something to be desired, with incorrect use of small brackets, and many grammatical errors. The contributions should also seriously be considered to be itemized for clarity. Provided that these presentation issues are fixed, I tentatively recommend acceptance.